# Population genomics and molecular epidemiology of wheat powdery mildew in Europe

Jigisha Jigisha[1], Jeanine Ly[1], Nikolaos Minadakis[1], Fabian Freund[2], Lukas Kunz[1], Urszula Piechota[3], Beyhan Akin[4], Virgilio Balmas[5], Roi Ben-David[6], Szilvia Bencze[7], Salim Bourras[8], Matteo Bozzoli[9], Otilia Cotuna[10], Gilles Couleaud[11], Mónika Cséplő[12], Paweł Czembor[3], Francesca Desiderio[13], Jost Dörnte[14], Antonín Dreiseitl[15], Angela Feechan[16,17], Agata Gadaleta[18], Kevin Gauthier[19], Angelica Giancaspro[18], Stefania L. Giove[18], Alain Handley-Cornillet[19], Amelia Hubbard[20], George Karaoglanidis[21], Steven Kildea[22], Emrah Koc[4], Žilvinas Liatukas[23], Marta S. Lopes[24], Fabio Mascher[25], Cathal McCabe[16], Thomas Miedaner[26], Fernando Martínez-Moreno[27], Charlotte F. Nellist[20], Sylwia Okoń[28], Coraline Praz[29], Javier Sánchez-Martín[30], Veronica Sărățeanu[10], Philipp Schulz[31], Nathalie Schwartz[11], Daniele Seghetta[32], Ignacio Solís Martel[27], Agrita Švarta[33], Stefanos Testempasis[21], Dolors Villegas[24,34], Victoria Widrig[30], Fabrizio Menardo[1]*

1 Department of Plant and Microbial Biology, University of Zurich, Zurich, Switzerland, 2 Department of Genetics, Genomics and Cancer Science, University of Leicester, Leicester, United Kingdom, 3 Plant Breeding and Acclimatization Institute - National Research Institute, Radzików, Poland, 4 CIMMYT-Turkey, Ankara, Turkey, 5 Department of Agricultural Sciences, University of Sassari, Sassari, Italy, 6 Department of Vegetable and Field Crops, Institute of Plant Sciences, Agricultural Research Organization–Volcani Institute, Rishon LeZion, Israel, 7 Hungarian Research Institute of Organic Agriculture, Budapest, Hungary, 8 Department of Plant Biology, Swedish University of Agricultural Sciences, Uppsala, Sweden, 9 Department of Agricultural and Food Sciences, University of Bologna, Bologna, Italy, 10 Agriculture Faculty, University of Life Sciences "King Mihai I" from Timișoara, Timișoara, Romania, 11 Arvalis Institut du végétal, Station Expérimentale, Boigneville, France, 12 Agricultural Institute, HUN-REN Centre for Agricultural Research, Martonvásár, Hungary, 13 Council for Agricultural Research and Economics, Research Centre for Genomics and Bioinformatics, Fiorenzuola d'Arda, Italy, 14 Deutsche Saatveredelung AG, Leutewitz, Germany, 15 Department of Integrated Plant Protection, Agrotest Fyto Ltd., Kroměříž, Czech Republic, 16 School of Agriculture and Food Science, University College Dublin, Belfield, Dublin, Ireland, 17 Institute for Life and Earth Sciences, School of Energy, Geosciences, Infrastructure and Society, Heriot-Watt University, Edinburgh, United Kingdom, 18 Department of Soil, Plant and Food Sciences, University of Bari Aldo Moro, Bari, Italy, 19 Agroscope, Department of Plant Breeding, Nyon, Switzerland, 20 NIAB Cambridge Crop Research, Cambridge, United Kingdom, 21 Department of Agriculture, Aristotle University of Thessaloniki, Thessaloniki, Greece, 22 Teagasc Crops Research, Oak Park, Carlow, Ireland, 23 Institute of Agriculture, Lithuanian Research Centre for Agriculture and Forestry, Akademija, Lithuania, 24 Sustainable Field Crops, IRTA, Lleida, Spain, 25 Haute école des sciences agronomiques, forestières et alimentaires, Bern, Switzerland, 26 State Plant Breeding Institute, University of Hohenheim, Stuttgart, Germany, 27 Department of Agronomy, University of Seville, Seville, Spain, 28 Institute of Genetics, Breeding and Biotechnology of Plants, University of Life Sciences in Lublin, Lublin, Poland, 29 Centro de Biotecnología y Genómica de Plantas, Universidad Politécnica de Madrid, Madrid, Spain, 30 Department of Microbiology and Genetics, Spanish-Portuguese Institute for Agricultural Research, University of Salamanca, Salamanca, Spain, 31 Institut für Pflanzenschutz in Ackerbau und Grünland, Julius Kühn-Institut, Bundesforschungsinstitut für Kulturpflanzen, Braunschweig, Germany, 32 Centro Ricerche e Sperimentazione per il Miglioramento Vegetale "N. Strampelli", Macerata, Italy, 33 Latvia University of Life sciences and technologies, Jelgava, Latvia, 34 Estacion Experimental de Aula Dei, CSIC, Zaragoza, Spain

* fabrizio.menardo@uzh.ch



**Data availability statement:** The authors confirm that all data underlying the findings are fully available without restriction. The short read genome sequence data generated from this study is available under the BioProject accession PRJEB75381. Accession numbers for all samples used in this study are available in S1 Data. The reference genome assembly as well as the VCF file used in this study are available at https://doi.org/10.5281/zeno-do.13903934. Code and data to reproduce all analyses is available at https://github.com/fmenardo/Bgt_popgen_Europe_2024/tree/Bgt_ms (archived at https://doi.org/10.5281/zenodo.15011360). All other data is contained within the manuscript and its supporting information.

**Funding:** This work was funded by the Swiss National Science Foundation (https://www.snf.ch/en) grant number PZ00P3_193473, awarded to FMe. The funders had no role in study design, data collection and analysis, decision to publish, or preparation of the manuscript.

**Competing interests:** The authors have declared that no competing interests exist.

**Abbreviations:** Bgt, Blumeria graminis forma specialis tritici; IBDe, identical-by-descent; LD, Linkage disequilibrium; PM, powdery mildew; WGS, whole genome sequencing.

## Abstract

Agricultural diseases are a major threat to sustainable food production. Yet, for many pathogens we know exceptionally little about their epidemiological and population dynamics, and this knowledge gap is slowing the development of efficient control strategies. Here we study the population genomics and molecular epidemiology of wheat powdery mildew, a disease caused by the biotrophic fungus *Blumeria graminis forma specialis tritici* (Bgt). We sampled Bgt across two consecutive years, 2022 and 2023, and compiled a genomic dataset of 415 Bgt isolates from 22 countries in Europe and surrounding regions. We identified a single epidemic unit in the north of Europe, consisting of a highly homogeneous population. Conversely, the south of Europe hosts smaller local populations which are less interconnected. In addition, we show that the population structure can be largely predicted by the prevalent wind patterns. We identified several loci that were under selection in the recent past, including fungicide targets and avirulence genes. Some of these loci are common between populations, while others are not, suggesting different local selective pressures. We reconstructed the evolutionary history of one of these loci, *AvrPm17*, coding for an effector recognized by the wheat receptor Pm17. We found evidence for a soft sweep on standing genetic variation. Multiple *AvrPm17* haplotypes, which can partially escape recognition by Pm17, spread rapidly throughout the continent upon its introduction in the early 2000s. We also identified a new virulent variant, which emerged more recently and can evade *Pm17* resistance altogether. Overall, we highlight the potential of genomic surveillance in resolving the evolutionary and epidemiological dynamics of agricultural pathogens, as well as in guiding control strategies.

## Introduction

Rapid improvement of DNA sequencing technologies accelerated the progress of molecular epidemiology in the last two decades. Large scale real-time whole genome sequencing (WGS) of pathogen populations has become essential for research, surveillance and monitoring of human diseases [1–3]. The most glaring example of this development was the global response to the COVID-19 pandemic, which at its peak saw the sequencing of hundreds of thousands of viral genomes each week. Among other things, this data has been used to track the epidemiological dynamics in different countries, follow the emergence and expansion of new variants, and design vaccines [4–6].

Beyond being a threat to human health, microbial pathogens are also a major issue in agriculture, where they cause an estimated loss of about 20% of some of the most important crops every year [7]. Long before the advent of whole genome sequencing, it was recognized that population genetics and molecular epidemiology studies could contribute towards understanding the biology of plant pathogens

and improving disease control [8–10]. The potential of these approaches, coupled with WGS, has renewed the interest in pathogen-informed strategies for pest management and resistance breeding [11–13]. Indeed, genomic data has been crucial to understand the emergence and dynamics of different agricultural diseases such as the olive quick decline syndrome in the south of Italy [14], the recent outbreaks of wheat blast in Zambia and Bangladesh [15,16], and the rapid shift in UK populations of wheat yellow rust [17].

Yet, despite significant progress, our knowledge about the population biology of most agricultural pathogens remains remarkably limited – How far can a pathogen disperse in one season? Which are the main directions and periods of dispersal? How many cycles of sexual and asexual reproduction occur each year? Where was the origin of the inoculum initiating a disease outbreak in a field? How connected are epidemics in different regions? – Often, there is but little evidence available to answer with confidence these and other basic questions about agricultural diseases.

In this study we focus on wheat powdery mildew in Europe and the Mediterranean, a region producing more than one-third of the global wheat harvest [18]. Powdery mildew is one of the most important wheat diseases, and it is caused by the ascomycete fungus *Blumeria graminis forma specialis tritici* (Bgt) [7,19,20]. Bgt is a host-specific obligatory biotroph infecting the epidermal tissue of wheat leaves, spikes, and stems. While it can occur everywhere wheat is cultivated, in Europe it is more prevalent at higher latitudes, as it favors temperate climates with cooler temperatures and high humidity [21]. As with other crop pathogens, powdery mildew is controlled with chemical (fungicides) treatments and by breeding resistant wheat varieties. However, Bgt populations are becoming less sensitive to fungicides [22–24], and new resistant host varieties have generally been effective for only a few years after their introduction [25–27]. The durability of genetic resistance depends both on its physiological and molecular mechanism, and on the evolutionary potential of the pathogen populations which are exposed to it [9,28,29]. While our knowledge of the molecular interactions between wheat and Bgt is improving rapidly [26,27,30–38], many fundamental aspects of the evolutionary and epidemiological dynamics of wheat powdery mildew are still unknown.

Though Bgt is known to reproduce both sexually and asexually, it was suggested that sexual reproduction events are rare based on the genomic analysis of four isolates [39]. It was also reported that European Bgt samples are genetically uniform and that there is no evidence for isolation by distance between them [40], suggesting that a single panmictic Bgt population is infecting wheat fields in the continent. However, these results were based on a handful of strains sampled in different decades. Another study based on the analysis of virulence spectra of wheat and barley powdery mildews concluded that North and Central Europe constitute a single epidemic unit [41]. It was also proposed that wheat powdery mildew spores are transported by the wind over two main axes, from south to north, following the progression of the vegetative season each year, and from west to east, following the prevailing wind direction [42,43]. But without molecular markers, many of these hypotheses could not be thoroughly tested. Virulence studies such as those mentioned above can identify and track "races" or "pathotypes" in pathogen populations. While they are valuable for breeding and to understand the evolution of virulence, they are based on phenotypic markers which are often under strong selective pressure, and are therefore of limited use to study population genetics [8,44].

Here, we overcome the limitations of previous studies with a dense and homogeneous sampling of Bgt populations during two consecutive years, 2022 and 2023, followed by WGS. We identified several distinct populations of wheat powdery mildew, highlighting a heterogenous landscape of gene flow. We found that the population structure and the gradients of diversity can be largely predicted by wind connectivity, suggesting that wind is an important factor shaping Bgt populations in Europe. We also discovered several loci that were under positive selection in the recent past and reconstructed the evolutionary history of one such locus, coding for the effector AvrPm17, which is recognized by the wheat resistance receptor Pm17. Overall, this study shows that (1) large-scale population genomics and molecular epidemiology have become feasible for agricultural pathogens, (2) they can generate insights about their basic biology, and (3) they can also provide valuable information for control strategies.

## Results

### Sampling and WGS

To study the nature, spread, and dynamics of wheat powdery mildew epidemics in Europe, we organized a large sampling effort over two seasons, 2022 and 2023, with the aim of achieving an unbiased and dense representation of Bgt populations. Overall, we collected 276 new Bgt strains from over 90 locations spread across 20 countries, spanning Europe and the neighboring Mediterranean region (see section "Methods", Fig A in S1 Text). We sequenced the haploid genome of all isolates with short reads, and we complemented this dataset with the publicly available genome sequences of 375 wheat powdery mildew isolates sampled from around the world between 1980 and 2019 (S1 Data) [26,33,36,40,45]. Short reads of the 651 isolates were mapped to the reference genome 96224 [46]. We excluded low-quality sequences and obtained 3,570,037 high-confidence biallelic SNPs from 568 samples (*World* dataset), which were used for downstream analyses (see section "Methods"). Principal component and ADMIXTURE [47] analyses of the *World* dataset showed that newly sampled isolates from Europe and the Middle East grouped closely with older samples from the same regions, and as reported previously [40], samples from other continents represented clearly distinct populations (Figs B–D in S1 Text, Appendix A in S1 Text).

### Population structure in Europe and the Mediterranean

Our sampling approach was designed to explore the population dynamics of wheat powdery mildew at a fine scale on a regional level. We defined a dataset of 415 isolates originating from Europe and surrounding regions (*Europe+* dataset; Fig A in S1 Text), and we investigated population structure with three methods: PCA, ADMIXTURE [47] and fineSTRUCTURE [48]. All analyses separated the Bgt isolates into different hierarchically clustered groups. The common groups that emerged from all analyses were composed of samples from different geographic regions, roughly corresponding to (i) Northern Europe, (ii) Southern Europe, (iii) Northern Turkey and Caucasus, (iv) Southern Turkey and Israel, and (v) Egypt (Figs D–F in S1 Text, Appendix A in S1 Text). In addition, fineSTRUCTURE resolved subtler population subdivisions which could not be detected by the other analyses (Fig G in S1 Text, Appendix A in S1 Text).

Overall, the population structure analysis of the *Europe+* dataset revealed that wheat powdery mildew does not constitute a single panmictic population in Europe and surrounding regions. Beyond the main subdivisions described above, we found that the Bgt population in Northern Europe covers a wide geographic range and is largely homogeneous, lacking strong substructure (Figs E–F in S1 Text). Conversely, in Southern Europe our results revealed two distinct sub-groups in Spain, and an additional one mostly composed of samples from Italy, suggesting lower gene flow, and local, less connected populations (Fig G in S1 Text).

### Classifying populations

While the analysis of population structure could distinguish different populations, the PCA, the ADMIXTURE results, and the fineSTRUCTURE coancestry matrix identified multiple "intermediate genotypes", which could not be clearly assigned to one group or the other (Figs D–F in S1 Text). In other contexts, these samples would be described as admixed. However, one limitation common to both ADMIXTURE and fineSTRUCTURE is that they model genetic diversity in terms of discrete ancestries or populations. Therefore, these might not be the best tools to investigate gradients of diversity, or continuous patterns of differentiation [49]. Moreover, it was shown before that patterns of shared ancestry can also be generated by sampling biases or demographic processes that do not include admixture [50]. Regardless, many population-level analyses require discrete populations. While we are aware that any subdivision is to some extent arbitrary, we chose the fineSTRUCTURE level-4 classification (see section "Methods", Fig H in S1 Text) that divided the dataset into five populations – **N_EUR** predominantly in Northern Europe, **S_EUR2** in Southern Europe, **TUR** in Turkey and Northern Caucasus, **ME** in the Middle East and **S_EUR1** comprising individuals that are geographically and genetically intermediate between N_EUR and S_EUR2 (Fig 1).

This classification was congruent with, and representative of, the main results of the PCA and of the ADMIXTURE analysis (Fig 1, Appendix A in S1 Text). All five populations showed a rapid rate of LD decay ($r^2$ < 0.2 within 1 Kb, Fig I in S1 Text). This, along with the finding that the two mating types occurred in similar proportions, indicates that sexual reproduction is pervasive in Europe (Table A in S1 Text). We calculated genome-wide distributions for several summary statistics using a sliding window approach. We found that the median nucleotide divergence between populations ($d_{xy}$) was between 0.0015 and 0.0017, while the median pairwise $F_{ST}$ was between –0.048 and 0.13, suggesting low to moderate levels of differentiation (Fig J in S1 Text). Finally, median values of within-population nucleotide diversity (pi) ranged between 0.0013 and 0.0016 (Fig 1e), and all populations had negative values of genome-wide Tajima's D, indicating an excess of rare variants (Fig K in S1 Text). For most downstream population-level analyses, we will focus on N_EUR and S_EUR2, as these were the most extensively sampled populations in 2022 and 2023 (Fig A in S1 Text).

**Demographic inference**

A genome-wide excess of rare variants is normally considered as evidence for population expansion. Indeed, a previous demographic analysis based on the Kingman coalescent inferred recent population growth for almost all Bgt populations [40]. Demographic inference based on the Kingman coalescent has become a fundamental tool in evolutionary biology and has been used for several plant pathogens [51,52]. However, one key assumption of this class of models is that the variation in reproductive success among lineages is small, and it was noted previously that this might not be the case for plant pathogens with seasonal epidemics [53]. Importantly, it was shown that when the assumption of small variance in reproductive success is violated, using the Kingman coalescent often leads to the inference of spurious population dynamics due to model misspecification. More specifically, populations characterized by large variance in reproductive success show an excess of rare variants, and demographic inference using the Kingman coalescent on such populations often results in the misleading inference of population growth [54,55].

We tested whether Bgt populations are characterized by a large variance in reproductive success, and whether the Kingman coalescent is an appropriate model to perform demographic inference of wheat powdery mildew. To exclude the confounding effect of population structure and of serial and non-random sampling, we focused on the four populations with the largest sample sizes identified by fineSTRUCTURE at the finest level of subdivision (level-10; Fig H in S1 Text) and only included isolates sampled in 2022, selecting one isolate from each location (see section "Methods" for details). We used the approach of Freund and colleagues [56] to test the fit of the Kingman coalescent and of the Beta coalescent, a model allowing for large variance in reproductive success. We found overwhelming support for the Beta coalescent in all four populations (likelihood ratio > 2,500), and evidence for no or minimal population growth (Table B in S1 Text). This was also true when we allowed for the possibility of confusing the derived and ancestral alleles (Table B in S1 Text). Moreover, the observed genome-wide site frequency spectrum (SFS) fitted very well to the expected SFS for the Beta, but not for the Kingman coalescent (Fig L in S1 Text).

These results indicate that the observed excess of rare variants in Bgt populations is due to the large variance in reproductive success, and not to population growth. However, this analysis cannot reveal the underlying biological process, and we will discuss some hypotheses further below. Finally, these findings show that the Kingman coalescent should not be used for demographic inference of Bgt, as it might lead to artifacts such as the inference of spurious population growth.

**Spatial population genetics**

The theoretical notion of discrete, randomly-mating populations is not ideal to study organisms that can travel large distances by wind. An alternative approach is to use methods that explicitly account for geography to explore the continuous distribution of genetic variation in space [57].

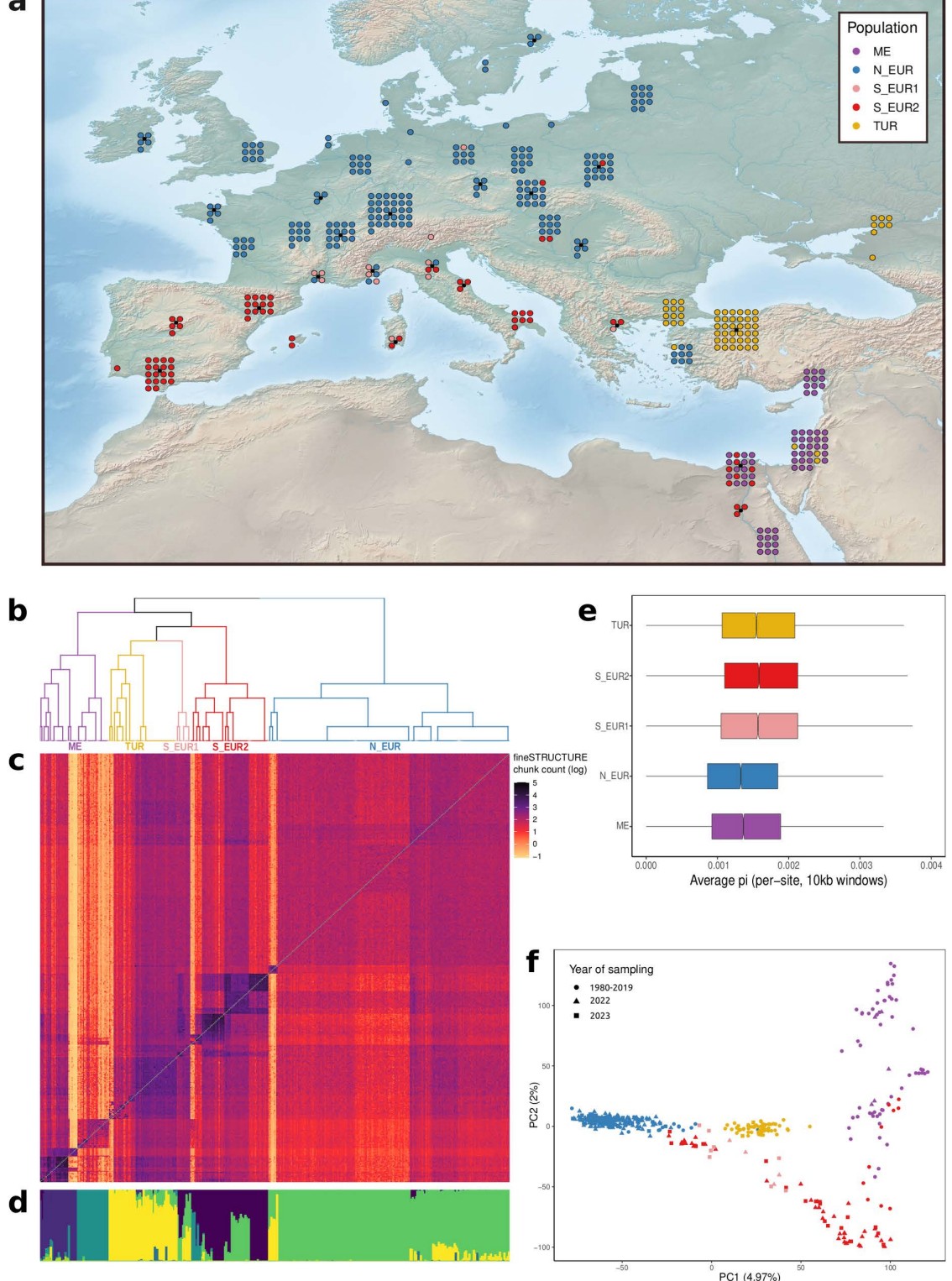

**Fig 1. Population structure of wheat powdery mildew in Europe and the Mediterranean. (a)** Map showing the geographical distribution of the five populations inferred from fineSTRUCTURE. The same color code is used in panels **a**, **b**, **e**, and **f**. **(b)** fineSTRUCTURE dendrogram representing similarity among populations; the five populations are labeled. **(c)** fineSTRUCTURE coancestry matrix. Chunk count is the number of genomic segments

donated by isolates in rows to isolates in columns (in log scale). Darker colors indicate higher shared ancestry. **(d)** Individual ancestry proportions of samples in the *Europe+* dataset based on ADMIXTURE's best run ($K = 9$). The analysis was performed on the *World* dataset, but only samples in the *Europe+* dataset are shown (see Appendix A in S1 Text for details). **(e)** Genome-wide average per-site nucleotide diversity (pi) of the five populations calculated in 10 kb windows. **(f)** Principal component analysis of the *Europe+* dataset. Each point represents an individual and the colors represent the populations as inferred from fineSTRUCTURE. The map was generated using a raster downloaded from Natural Earth (https://www.naturalearthdata.com/downloads/50m-raster-data/). The data underlying this figure can be found in https://doi.org/10.5281/zenodo.15011360.

To investigate spatial gradients of genetic diversity, we focused on the *Europe+_2022_2023* dataset, which comprises samples collected at roughly the same time (2022–2023) from locations distributed nearly uniformly over Europe (Fig A in S1 Text). We found a strong spatial structure, consistent with isolation by distance (Table C in S1 Text), i.e., individuals in geographic proximity were also genetically more similar (Fig 2; Mantel test correlation = 0.472, *p*-value = 0.001). However, the bimodal distribution of pairwise genetic distances (Fig 2c, Fig M in S1 Text) suggested the presence of a discrete underlying population structure in addition to continuous clines of differentiation. We therefore repeated the analysis individually within the two main populations in the *Europe+_2022_2023* dataset, N_EUR and S_EUR2 (Fig 2a-b). For both populations, the distribution of genetic distances was unimodal (Fig M in S1 Text), and we found a weaker signal of isolation-by-distance in N_EUR (Mantel test correlation = 0.085, *p*-value = 0.011) compared to S_EUR2 (Mantel test correlation = 0.389, *p*-value = 0.001).

This pattern of heterogeneity in isolation-by-distance was also captured by the effective migration surfaces estimated using FEEMS [58,59]. Briefly, this method infers the magnitude of effective migration in a certain region based on the rate of decay of genetic similarity. Northern Europe was characterized as a uniform area of higher-than-average effective migration, while in Southern Europe there was much lower effective migration overall, and a more heterogeneous landscape of gene flow (Fig 2d). These findings complement the results of the previous population structure analyses described above. The high rate of gene flow in the north is reflected in the large homogenous Bgt population of Northern Europe, while lower gene flow in the south is responsible for smaller local populations that are not as well-connected (Figs 1 and 2).

### Factors shaping genetic diversity

Wind is considered as the main dispersal agent of powdery mildew [43]. Conidiospores are the clonal propagules of Bgt, and it was shown that they can be transported by wind for hundreds of kilometers before landing on a new host [42]. Here, we tested whether wind connectivity, or other environmental variables, can explain the observed patterns of diversity, both between and within populations. We used 'windscape' [60,61] to estimate wind-distances between each pair of sampling locations. These can be interpreted as the average time needed by air masses to move between two points, so that locations well connected by wind have small wind-distances, and vice versa. Additionally, we estimated pairwise climatic distances, which quantify differences in the climates of distinct locations (see section "Methods").

We tested whether population structure can be explained by wind, climatic, or geographic distances. We focused on the two main populations in the 2022–2023 dataset (N_EUR and S_EUR2) and used logistic regression to model the probability of two isolates being classified in the same or different populations based on wind, climate, and geographic distances. Both simple and multiple logistic regression showed that wind distance was the best predictor of population structure ($R^2 = 0.497$ in the simple model), and adding climatic and geographic distances to the model increased its explanatory power only marginally ($R^2 = 0.533$ in the full model; Table E in S1 Text).

We also tested how well wind distances correlated with genetic distances (isolation by wind distance) and compared these results with isolation by geographic and climatic distances. Importantly, these three measures are highly correlated, as distant locations tend to be less connected by winds and have different climates (Table D in S1 Text). We found that wind-distances had the highest correlation with genetic distances in the N_EUR population and overall, while geographic and climatic distances had a higher correlation with genetic distances in S_EUR2 (Table C in S1 Text, Fig N in S1 Text).

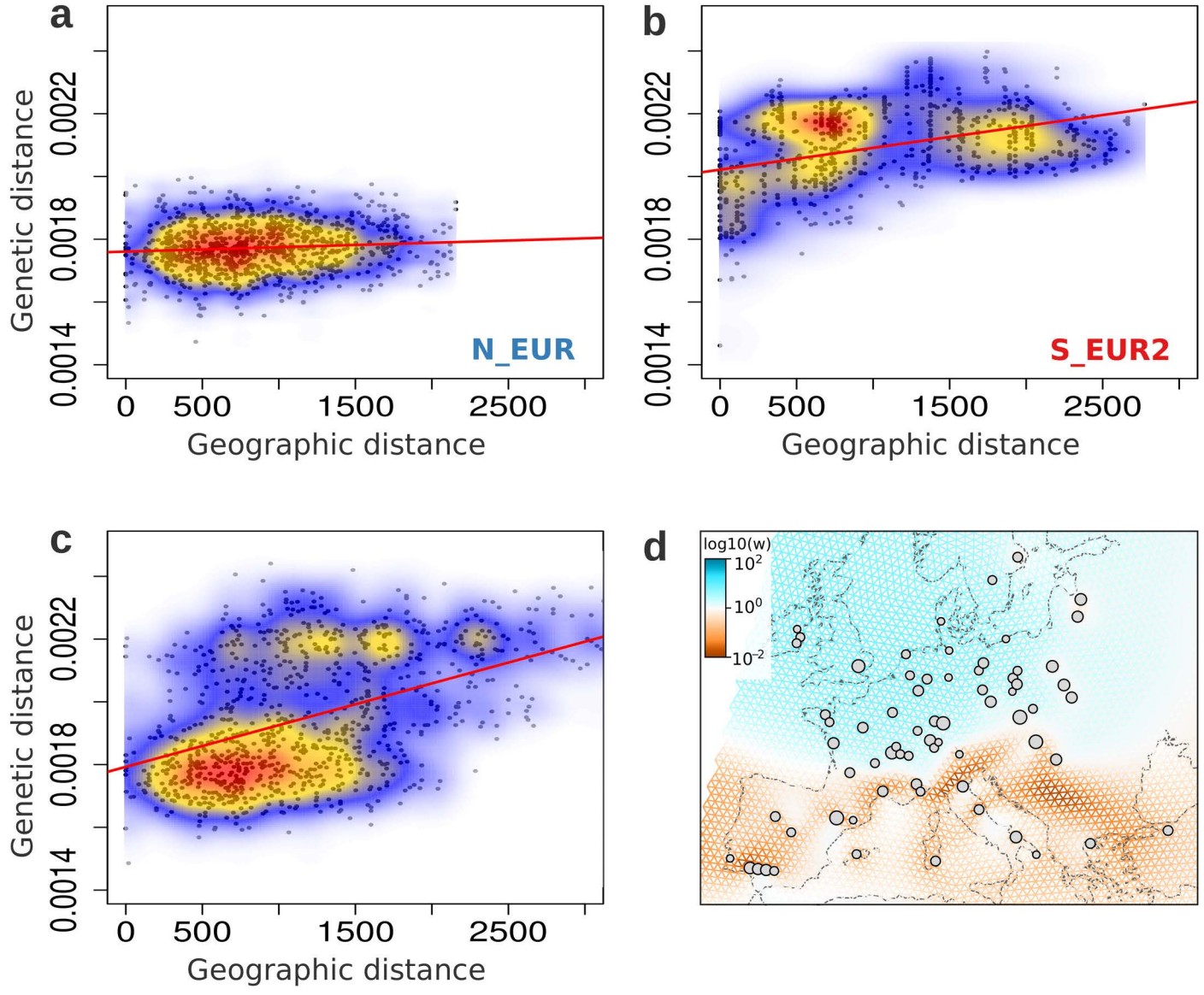

**Fig 2. Spatial population structure of wheat powdery mildew in Europe.** Isolation-by-distance in the *Europe+_2022_2023* samples belonging to the **(a)** N_EUR population, **(b)** S_EUR2 population, and **(c)** the complete *Europe+_2022_2023* dataset. Genetic distance is the number SNPs between a pair of individuals, scaled by the total number of loci compared. Geographic distance is measured between sampling locations of pairs of individuals, in kilometers. The colors represent the density of the data points, with warmer colors corresponding to higher density. 1,000 randomly sampled data points for each dataset are plotted in black. The Mantel test correlation values for **a, b**, and **c** are 0.085*, 0.389*** and 0.472***, respectively. (*$p \leq 0.05$, ***$p \leq 0.001$) **(d)** Effective migration surfaces estimated using FEEMS for the *Europe+_2022_2023* dataset (smoothness parameter lambda = 2.06914). The colors indicate the inferred relative intensity of gene flow along each edge of the spatial graph. Orange corresponds to lower-than-average effective migration and blue, higher-than-average. The grey circles show the nodes of the spatial grid the samples were assigned to, based on their sampling locations. The size of the node is proportional to the number of samples assigned. The analysis was performed on the complete *Europe+_2022_2023* dataset, but for graphic reasons, the map was cropped and isolates from Israel are not shown. The map was generated using a land shapefile from Natural Earth (https://www.naturalearthdata.com/features/). The data underlying this figure can be found in https://doi.org/10.5281/zenodo.15011360.

To gain additional insights, we qualitatively examined both inbound and outbound wind connectivity surfaces with respect to several locations in Europe to identify predominant wind patterns (Fig 3a-b, Fig O in S1 Text). This analysis confirmed that Northern Europe is well connected by winds, which flow mainly along the east-west axis, with a primary

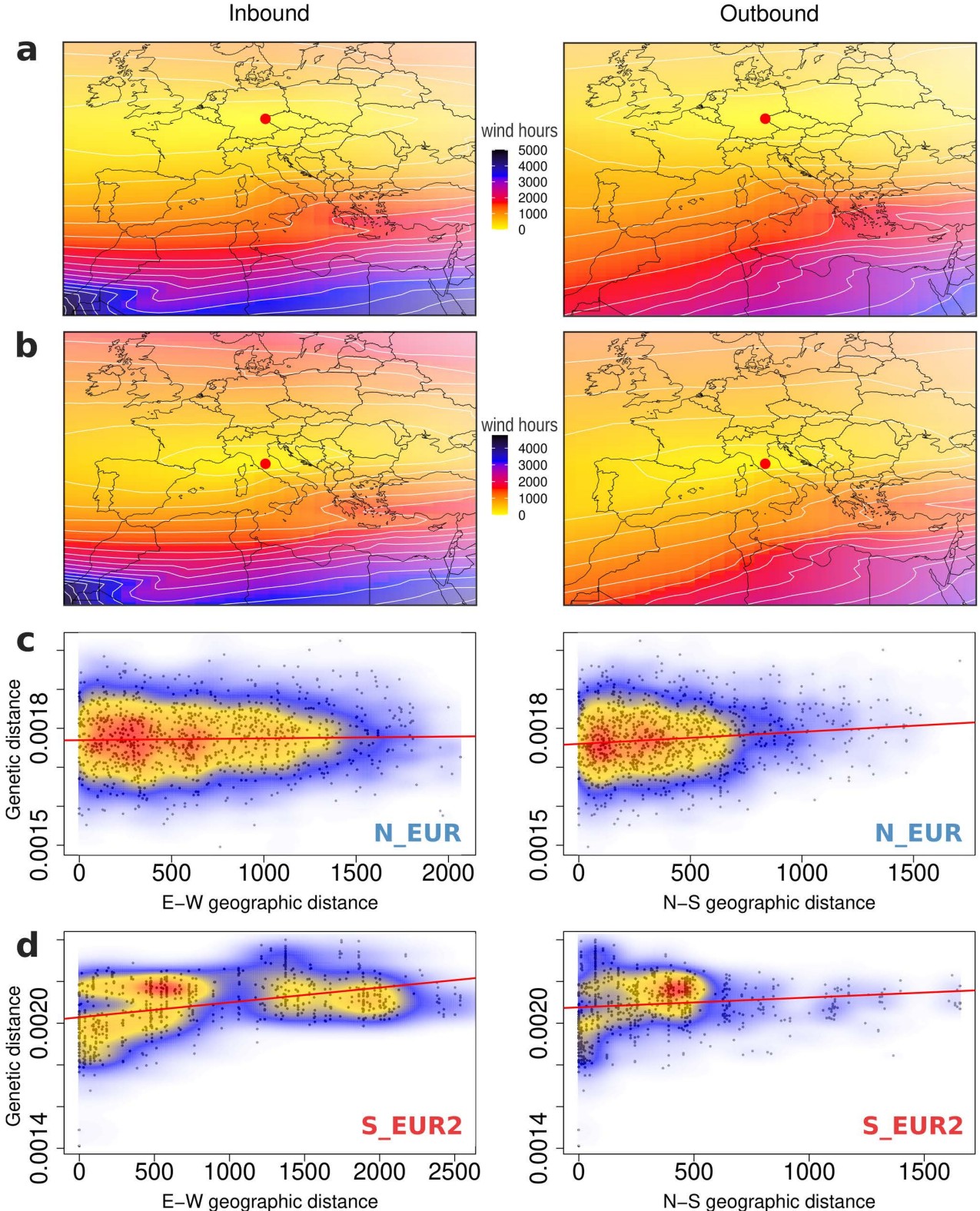

**Fig 3. Patterns of wind connectivity and isolation by distance.** Inbound and outbound wind connectivity surfaces based on wind data from 2012 to 2021 with respect to focal points (red circles) in (**a**) Northern and (**b**) Southern Europe. Wind connectivity, measured in wind hours (see section

"Methods"), represents how long it takes for air mass to move to or from the focal location. Lighter colors imply higher wind connectivity. **(c)** Isolation by distance measured separately along the east-west and north-south geographic axes (columns 1 and 2, respectively) for isolates sampled in 2022 and 2023 belonging to the N_EUR population and **(d)** to the S_EUR2 population. Genetic distance is the number of SNPs between a pair of individuals scaled by the total number of loci compared. Geographic distance along the two axes is measured between sampling locations of pairs of individuals, in kilometers (see section "Methods" for details). The colors represent the density of the data points, with warmer colors corresponding to higher density. 1,000 randomly sampled data points for each dataset are plotted in black. Results from the corresponding Mantel tests are reported in Table F in S1 Text. The maps were generated using a shapefile from the R library mapdata (https://cran.r-project.org/web/packages/mapdata/index.html). The data underlying this figure can be found in https://doi.org/10.5281/zenodo.15011360.

direction from west to east (Fig 3a). We tested if this was also reflected in how rapidly genetic similarity decayed over the north-south and east-west axes. Indeed, we found isolation-by-distance on the north-south axis, but not along the east-west axis in the N_EUR population (Table F in S1 Text, Fig 3c). In Southern Europe, the wind connectivity surfaces were qualitatively similar to those in the north (Fig 3b, Fig O in S1 Text). However, the S_EUR2 population exhibited a stronger isolation-by-distance signal along the east-west axis compared to the north-south axis, probably because the sea hinders gene flow on the east-west axis (Table F in S1 Text, Fig 3d).

To further disentangle the effect of different variables on genetic diversity, we performed redundancy analysis (RDA). We found that geography, wind and climate all had significant effects (Table G in S1 Text). In addition, the host of collection (hexaploid versus tetraploid wheat) was also a significant predictor of genetic variation, suggesting a degree of host specialization on bread and durum wheat (Table H in S1 Text, Appendix B in S1 Text).

Overall, these results indicate that several factors are likely to shape the genetic variability in European Bgt populations, and that wind connectivity appears to be a particularly good predictor of the subdivision between Northern and Southern Europe.

### Spatiotemporal patterns of genetic variation

We used samples collected in different periods to explore the spatial epidemiology of wheat powdery mildew at different timescales. The sampling from two consecutive seasons (2022 and 2023) enabled investigation at a finer temporal scale compared to what can be achieved with classic population genetics. The older collection of isolates from the 1980s and 1990s allowed us to make comparisons with contemporary populations and examine long-term patterns.

To understand how Bgt populations changed from one year to the next, we used the finest level of population subdivision inferred by fineSTRUCTURE (level-10, Fig H in S1 Text). At this level, fineSTRUCTURE distinguished 45 populations, each of them composed of "statistically identical" individuals (Appendix A in S1 Text). We focused on 14 of these populations (4 in Northern Europe, 10 in Southern Europe) that included 96% of the isolates sampled in 2022 and 2023 (244 of 255). We found that each of these populations contained samples collected in both years from nearby locations (Fig P in S1 Text). In other words, we could not detect a geographic shift of any population from 2022 to 2023, and for most locations, we sampled individuals from populations that persisted locally for the two seasons, a result that was also corroborated by the PCA (Fig E in S1 Text).

The four populations from Northern Europe were characterized by a wider geographic distribution, as discussed previously. For example, most isolates sampled from Northwestern Europe were grouped into the same population N_EUR2, independent of the year of sampling (Fig P in S1 Text). Similar patterns were observed for the other Northern European populations N_EUR1, E_EUR1 and E_EUR2 (Fig P in S1 Text). The smaller populations of Southern Europe also persisted locally from 2022 to 2023. For instance, samples collected from Catalonia (Northeast Spain) in 2022 were either grouped with samples from Central Spain in the SPAIN_N2 population or classified into a separate population SPAIN_N3. All Catalonian samples from 2023 also fell into one of these two populations (Fig P in S1 Text). Likewise, for several locations in Italy, Southern Spain and Southern France, we sampled from the same pools of genetic diversity in 2022 and 2023 (Fig P in S1 Text).

One specific hypothesis we aimed to test with this analysis was that advanced by Limpert and colleagues [43], proposing that one of the main directions of dispersal for Bgt is from west to east due to the prevailing wind direction. However, with samples from two consecutive years we could not observe a geographic shift in any direction. To test if this hypothesis held true over longer time frames, we leveraged the older collection of Bgt from Northwestern Europe. All except one of the isolates sampled before the 2000s from Northern Europe belonged to one of two closely related populations: N_EUR_old and N_EUR_old+ (fineSTRUCTURE level-10, Fig Q in S1 Text). Within these two populations comprising 20 isolates, we found only three that had been collected after the year 2000. We used the fineSTRUCTURE dendrogram to identify which populations were most similar to these two, and we found them to be two populations comprising exclusively of recent isolates (sampled between 2018 and 2023) from Eastern Europe (E_EUR1 and E_EUR2) (Fig Q in S1 Text), suggesting a west to east movement of Bgt populations over a period of 20–30 years.

Overall, with this analysis we showed that populations persisted locally between 2022 and 2023, while over a longer period of time, populations in Northern Europe appear to be moving eastwards. However, more data is needed to confirm these results.

## Signatures of recent selection in Bgt populations

Bgt populations in different regions are exposed to different environments, wheat genotypes, and agricultural practices. Such diversity of conditions can result in distinct selective pressures acting on local populations. We explored this using genome scans for selection based on the detection of "identical-by-descent" (IBDe) segments between pairs of isolates. More specifically, we used isoRelate [62] to infer IBDe segments between all pairs of isolates within each of the five populations described above (Fig 1). IBDe segments inferred by isoRelate are large contiguous stretches of chromosomes (at least 2 cM and 50 Kb), that are (nearly) identical between two strains because they were inherited from a common ancestor approximately within the last 25 sexual generations. Chromosomal regions that show a significant excess of relatedness (i.e., an excess of IBDe pairs) compared to the rest of the genome represent loci that, over several generations, have been inherited by more offsprings than expected by chance, indicating that at least one of the alleles at such loci conferred a fitness advantage. In other words, this analysis detects recent selective sweeps.

We found several loci putatively under recent positive selection (Fig 4, Figs R–S in S1 Text). For example, *cyp51* (14α-sterol demethylase), a gene encoding for the target of demethylation inhibitor fungicides, shows a strong signature of recent selection in all populations except the Middle East (ME) [63]. Furthermore, the genomic region containing *AvrPm17*, an avirulence gene coding for an effector recognized by the wheat resistance receptor Pm17, showed an excess of relatedness in Southern Europe and the Middle East (S_EUR1, S_EUR2, and ME), but not in Northern Europe and Turkey (N_EUR, TUR).

This analysis revealed the landscape of selective pressures that acted on different Bgt populations in the last few decades. While it was not possible to identify the selective pressure responsible for each peak in relatedness, the cross-matching of such peaks with known avirulence genes and fungicides targets showed that the selective pressures imposed by fungicide treatments and resistance breeding were among the most important forces acting on wheat powdery mildew in the last decades (Fig 4, Fig R in S1 Text).

## The recent evolutionary history of *AvrPm17*

One example of a recent source of selective pressure affecting wheat powdery mildew is the resistance gene *Pm17*. *Pm17* was introduced into the hexaploid wheat gene pool through a translocation from rye (1AL.1RS) which was deployed in Europe at the beginning of the 21st century. However, its effectiveness proved short-lived, and Bgt populations in Europe and other continents have largely overcome the resistance provided by 1AL.1RS (Appendix C in S1 Text; [26,64]).

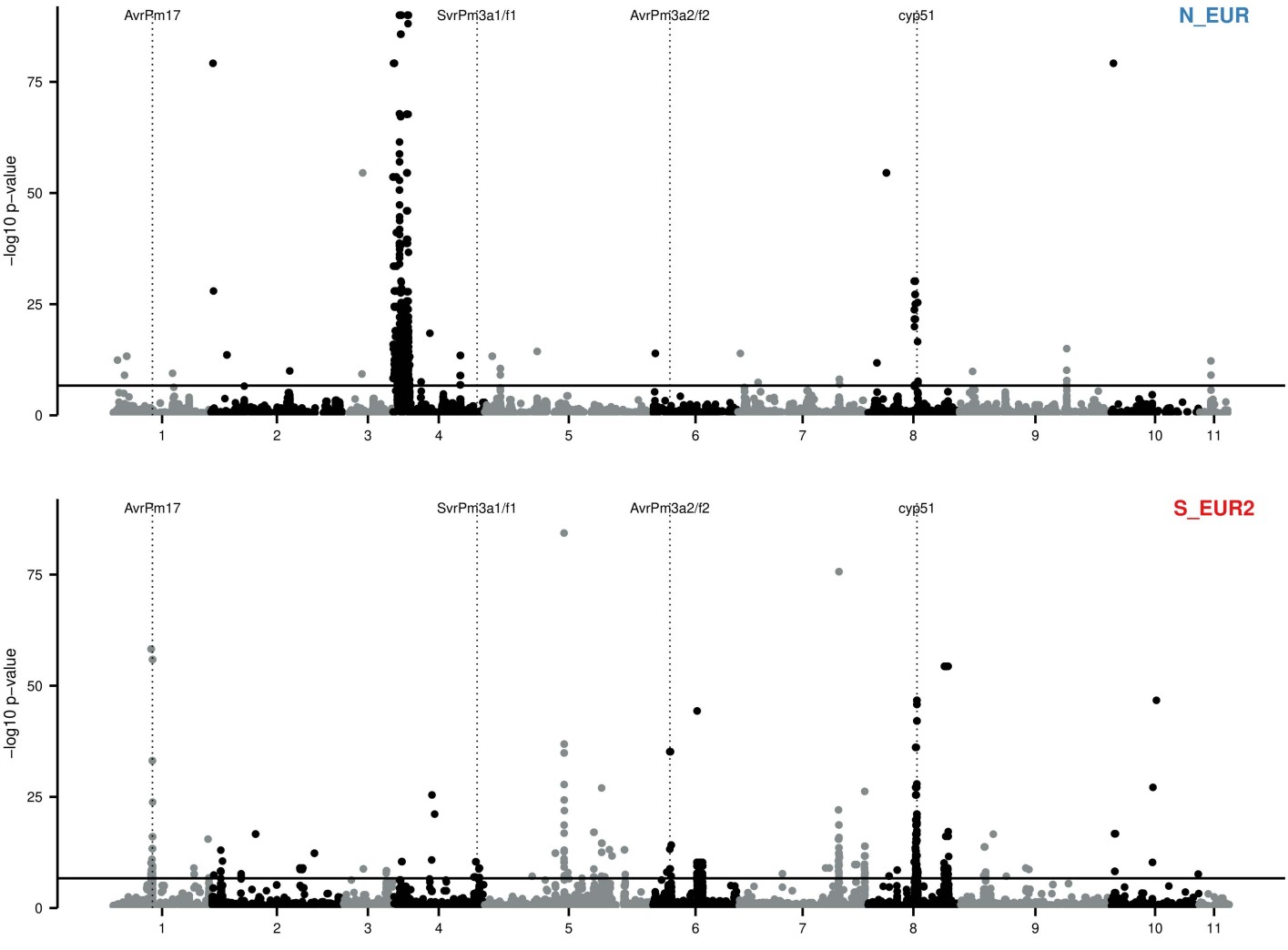

**Fig 4. Genome-wide scans for signatures of recent selection.** Manhattan plot for the isoRelate analysis of populations N_EUR and S_EUR2. SNPs in region with a significant excess of identical-by-descent pairs indicate loci under positive selection in the recent past (approximately 25 sexual generations). The horizontal full lines show the Bonferroni corrected 0.05 threshold. Vertical dotted lines show the position of two known avirulence genes (*AvrPm17* and *AvrPm3a2/f2*), one suppressor of virulence (*SvrPm3a1/f1*), and the fungicide target *cyp51*. SNPs with –log *p*-values <0.5 are not shown. The data underlying this figure can be found in https://doi.org/10.5281/zenodo.15011360.

To understand how resistance to *Pm17* originated and spread in Europe, we investigated the molecular epidemiology and population genetics of *AvrPm17*, the avirulence gene coding for the effector recognized by Pm17 [26]. We found that all isolates harbored at least one copy of *AvrPm17*, suggesting that this gene has an important virulence function, and that gene loss is not a viable pathway to overcome *Pm17*. We also found that the number of *AvrPm17* copies in each genome is variable, with nearly 90% of the isolates harboring two copies (Fig T in S1 Text, Appendix C in S1 Text). We identified three main protein variants which were characterized previously: variants A, B, and C, and six additional ones, namely F, H, I, J, K and L (Table I in S1 Text, Fig 5a-b). It was shown previously that variant A triggers a strong immune response when recognized by Pm17, and isolates with this variant are avirulent on *Pm17* transgenic wheat lines. Conversely, variants B and C elicit a weaker reaction, and they are partially virulent on *Pm17* lines [26]. Among the novel variants, H, which differs from variant C by one amino acid change, was found in 12 isolates, while the remaining five variants were found at most in two samples (Table I in S1 Text, Fig 5b, Fig U in S1 Text, Appendix C in S1 Text). We found that B and C

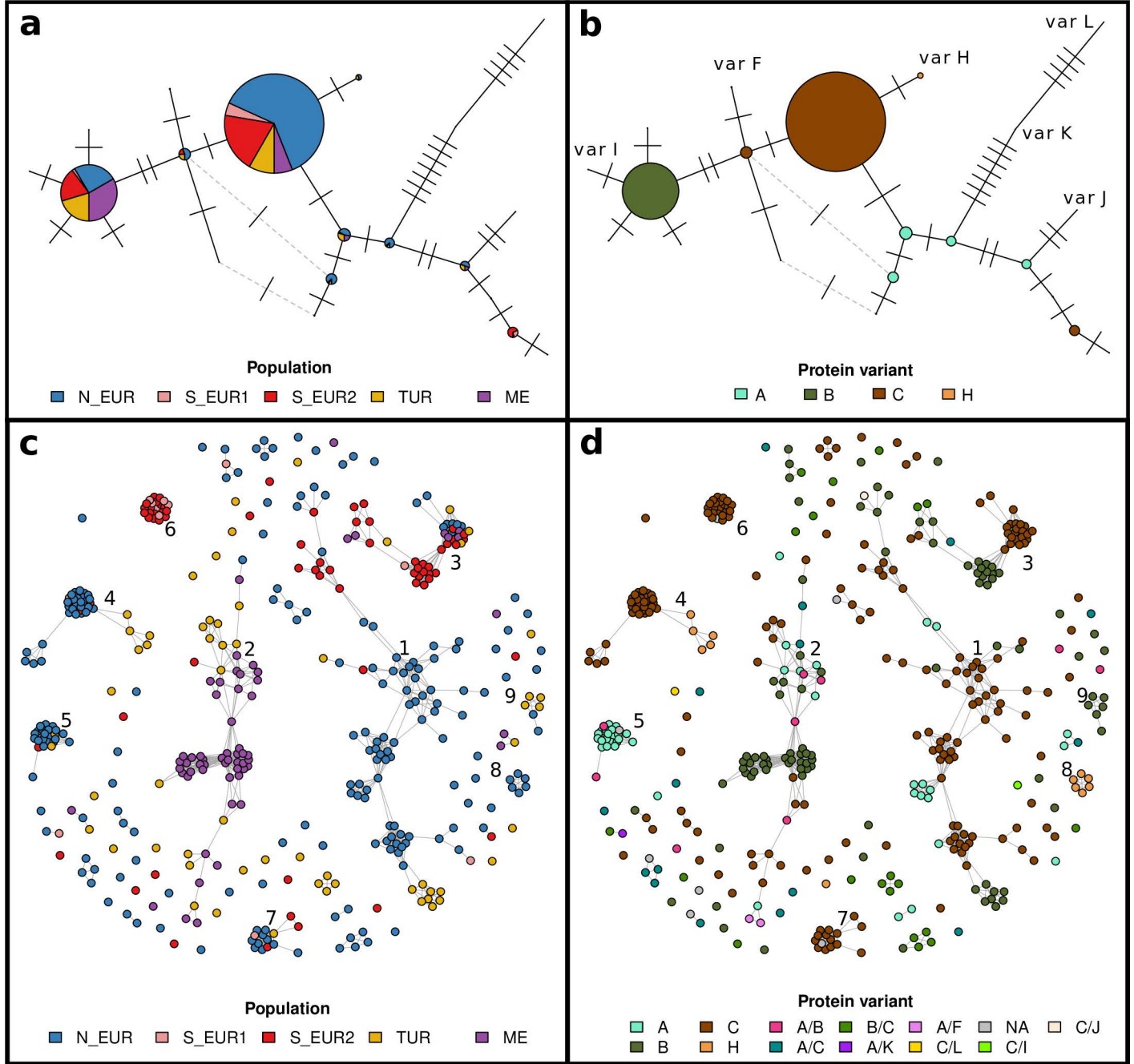

**Fig 5. Relatedness and haplotype networks of *AvrPm17*. (a)** Haplotype network of *AvrPm17* with haplotypes colored by population. Each node of the network represents a unique nucleotide sequence (haplotype). The size of nodes is proportional to the number of isolates in which that haplotype was observed. Ticks on edges connecting different nodes represent the number of nucleotide differences between two haplotypes. Dashed grey edges represent alternative connections. Alternative connections connecting haplotypes with more than two nucleotide differences are not plotted. **(b)** Haplotype network of *AvrPm17* with haplotypes colored by protein variant. The network is identical to **a**, but haplotypes are colored based on their mature amino acid sequence (after cleavage of signal peptide). Rare protein variants are not color coded but their position on the network is labelled. **(c)** Relatedness network for the *AvrPm17* locus. Each node represents one isolate (415 samples belonging to the *Europe+* dataset), edges connect isolates that are identical-by-descent over the *AvrPm17* locus. Isolates are colored by population, and the nine clusters with more than five samples are labelled. **(d)** Same relatedness network as in **c**. Isolates are colored based on the protein variant(s) coded by their respective *AvrPm17* genes. The data underlying this figure can be found in https://doi.org/10.5281/zenodo.15011360.

were the most frequent variants and were present respectively in 125 and 246 of the 415 Bgt isolates in the *Europe+* dataset (Table I in S1 Text). Conversely, A was found only in 70 samples, and its frequency significantly decreased over the last decades (Table J in S1 Text, Fig U in S1 Text, Appendix C in S1 Text).

We identified all pairs of isolates that were identical-by-descent (IBDe) over the *AvrPm17* locus, and we built a relatedness network in which strains were clustered together if they inherited the *AvrPm17* locus from the same recent common ancestor (Fig 5c-d; [62]). The relatedness network revealed nine clusters with six or more isolates, indicating that multiple unrelated *AvrPm17* haplotypes are circulating in the continent. Moreover, variant C was strongly associated with large clusters, suggesting a fitness advantage (Table K in S1 Text). This pattern is consistent with a soft sweep on standing variation upon the introduction of *Pm17* in Europe, and indeed the three most common protein variants were already present in European populations before *Pm17* was deployed (Fig U in S1 Text, Appendix C in S1 Text). We tracked the geographic spread of the major IBDe clusters (Figs V–W in S1 Text) and found that some *AvrPm17* haplotypes were dispersed over the whole continent and were transferred between genetically distinct populations (e.g., clusters 1 and 3; Fig V in S1 Text). Others were confined to a limited region and consisted of isolates belonging mostly to one or few populations (e.g., clusters 4–8; Figs V–W in S1 Text). Some of these are younger clusters; for example, cluster 8 comprises five isolates carrying variant H. They share long IBDe segments, pointing to a very recent common ancestor for *AvrPm17* (Fig X in S1 Text).

Variant H was only observed in Northern Europe and Turkey in samples collected in 2017 or later, and these isolates belonged to two distinct IBDe clusters, indicating two independent origins (Fig 5d). Interestingly, variant H evolved from variant C through one amino acid mutation (Y31H), and we found the same mutation in a variant B background, generating variant I (Fig 5a-b, Table I in S1 Text). The independent emergence of Y31H in different clusters and haplotypes suggests that it might be a beneficial mutation. We tested this hypothesis by repeating the functional validation assays used in a previous study to characterize *AvrPm17* [26]. Indeed, we found that variant H was not recognized by Pm17 when the two proteins were co-expressed in *Nicotiana benthamiana*. Moreover, isolates carrying variant H were fully virulent on *Pm17* transgenic wheat lines (Fig 6, Figs Y–Z in S1 Text). These results demonstrate that the amino acid mutation Y31H enables variant H to completely evade recognition by Pm17. While variant H is still rare in the European population (3% in the *Europe+_recent* dataset), it provides a fitness advantage on *Pm17* lines, and it might expand in the future depending on the strength of the selective pressure imposed by *Pm17*, and on its fitness cost on wheat lines without this resistance gene.

## Assessing the potential of breeding lines with real-time pathogen collections

The findings presented above suggest that the durability of novel resistance could be predicted with information about real-time pathogen populations. Specifically, had we known that AvrPm17 variants capable of partially escaping recognition by Pm17 were already present in Europe when *Pm17* was deployed, we could have anticipated the rapid breakdown of resistance. Large, (almost) real-time collections can provide valuable information in this context. As proof of concept, we consider the resistance provided by a different (powdery mildew) *Pm* gene, *Pm3e*, which has not been used in breeding programs, but was identified as a promising candidate by some recent studies [65,66]. *Pm3e* transgenic lines were tested in a field trial at one location in Switzerland for nine seasons, in which they consistently proved to be immune to powdery mildew. We leveraged our collection of European isolates sampled in 2022 to test their virulence on one of the *Pm3e* transgenic lines that underwent field trials. We found that seven isolates (5% of the tested ones) were at least partially virulent, and three of them, sampled in Switzerland, Germany, and Sweden, have completely overcome *Pm3e* resistance in our seedling infection assay (Fig AA in S1 Text).

These results show that a pool of virulent genotypes is already present in the European population of Bgt. In the hypothetical case in which this transgenic *Pm3e* line was to be cultivated on a large scale, and assuming no major fitness costs associated with virulence on *Pm3e*, it is likely that those genotypes would rapidly increase in frequency and spread

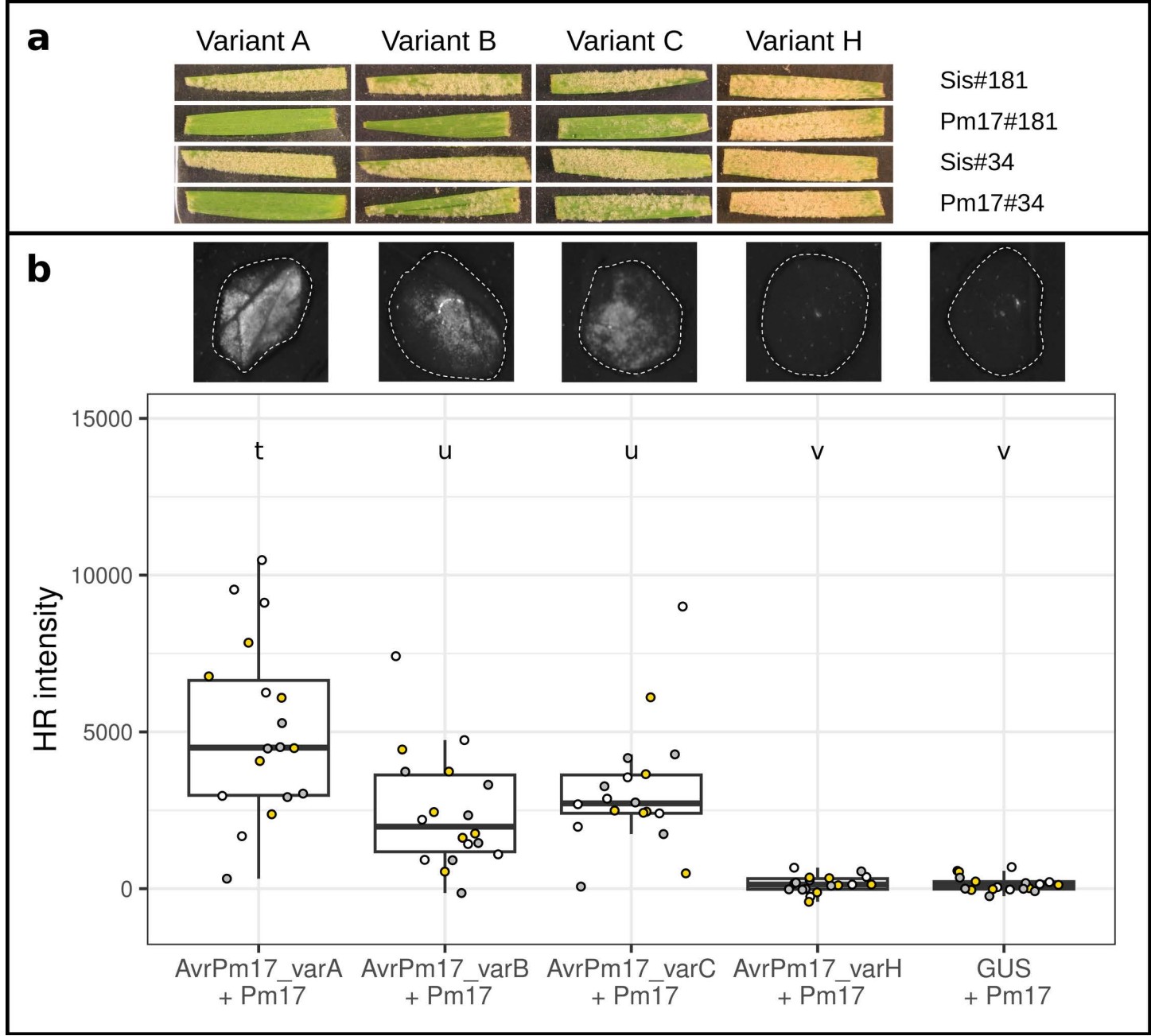

**Fig 6. Variant H is not recognized by Pm17. (a)** Results of the infection assay on two transgenic wheat lines containing *Pm17* (Pm17#181 and Pm17#34), and the two sister lines as controls (Sis#181 and Sis#34). The leaf fragments were infected with isolates carrying the four different variants. We tested two isolates for each of the four variants A, B, C and H. In this figure we included one representative leaf fragment per variant and wheat line (Variant A: DEBE032203; Variant B: ESAN042315; Variant C: DKES062203; Variant H: DEHA062201). Full pictures for all infection tests are reported in Figs Y–Z in S1 Text. **(b)** *Agrobacterium*-mediated co-expression of AvrPm17 variants with Pm17 in *N. benthamiana*. Co-expression of GUS + Pm17 serves as a negative control. Co-infiltrations were performed at a 4 (*AvrPm17*): 1 (*Pm17*) ratio with *n* = 6 leaves and repeated independently a total of three times (total *n* = 18 leaves). Leaves were imaged at two days post inoculation using a Fusion FX imager system. Infiltration pictures for one leaf are shown above the boxplot for each variant (same order as the infection tests) and for the negative control. Datapoints are color coded to represent the three independent experiments. Letters above the boxplots represent statistical differences ($p < 0.05$) as determined by a pairwise Wilcoxon rank sum exact test. The data underlying this figure can be found in https://doi.org/10.5281/zenodo.15011360.

throughout the continent. Similar to what was observed for *Pm17*, this new source of resistance would also quickly become ineffective.

## Discussion

Our understanding of the molecular interactions between plants and their pathogens has improved greatly in the last years. For wheat, it was predicted that progress will likely accelerate rapidly in the future, and that the majority of known major resistance genes will be cloned before the end of the next decade [28]. Together with the deployment of quantitative resistance, the bioengineering of wheat receptors and gene pyramiding may soon become viable pathways to achieve resistance with longer durability [67–71]. On the pathogen side, the catalog of effectors recognized by the wheat immune system is expanding rapidly, in Bgt and in other (hemi-) biotrophic pathogens [26,27,30,31,33,34,36,37,72,73].

To take full advantage of these resources, more information about natural pathogen populations is needed. Recent studies using large global collections of wheat pathogens and their whole-genome sequences have produced valuable insights into their evolutionary history and global diversity [16,26,40,74–76]. However, the resolution of these studies was often insufficient to resolve fine-scale evolutionary and epidemiological dynamics (but see ref. [76]). For instance, a lack of population subdivision was reported within Europe for Bgt and *Zymoseptoria tritici*, but this may be because many regions of the continent were underrepresented [40,75]. Here we reported the results of a fine-scale genomic surveillance study repeated for two consecutive seasons in Europe, and we showed that such data can reveal detailed spatial and temporal epidemiological dynamics.

### The population biology of wheat powdery mildew

A critical finding of this study is that powdery mildew epidemics in different regions of Northern Europe are caused by one single population, which is maintained genetically homogeneous over large distances by wind dispersal (Figs 1-3). This supports the hypothesis of Limpert and colleagues [43], who proposed the existence of a single epidemic unit over Northern Europe based on changes in virulence over time. Conversely, in Southern Europe local populations are less connected, probably due to a more fragmented habitat for the pathogen. In this region land masses are separated by large sea stretches, and wheat growing areas are rare and more isolated [77]. The combination of host availability and wind connectivity can also explain the diverging pattern of isolation-by-distance and isolation-by-wind within populations (Table C in S1 Text, Fig N in S1 Text). We found a strong isolation-by-wind signal in Northern Europe, while in the south, wind and genetic distances were not as strongly correlated. This is not surprising considering that the main winds blow predominantly from west to east, and that both the Iberian and Italian peninsula are oriented on the north-south axis. Thus, the homogenizing action of the wind is lost in Southern Europe, due to the seas that constitute a barrier to gene flow between these two land masses. Furthermore, the results of the population structure analyses and of FEEMS suggest that the two main European mountain ranges, the Alps and Pyrenees, also constitute natural barriers to dispersal, likely due to the lack of host, and to reduced wind connectivity between the two sides of these mountain ranges.

Previous studies, based on the analyses of virulence spectra, hypothesized that spore dispersal occurs mainly along the south-north axis following host vegetative development within a season (the green wave hypothesis), and the west-east axis following wind patterns [43]. Our data rejects the green wave hypothesis, at least at the continental level (some similar process could occur locally). At the same time, we found evidence for an eastward shift of Bgt populations over time (Fig Q in S1 Text). However, this analysis was limited by the low sample size and reduced geographic range of older collections. Asymmetric gene flow from west to east was reported for the wheat pathogen septoria tritici blotch (*Zymoseptoria tritici* [78]) and is common for wind-dispersed organisms [61]. Increased sampling efforts will help clarify this hypothesis while also uncovering other spatial dynamics.

Our population genetics analysis revealed that gene flow is limited between different regions in Europe, and that this is caused mainly by a combination of low wind connectivity and host availability. In addition, we identified distinct signatures

of selection for each population, showing that despite the high dispersal ability, Bgt populations might be adapted to local environmental conditions (in a broad sense). Indeed, climatic conditions and the genetic background of the host (hexaploid versus tetraploid) on which each sample was collected explained a part of the genetic variability in our dataset (Tables G–H in S1 Text, Appendix B in S1 Text). Beyond the differences between tetraploid and hexaploid wheat, it was reported that hexaploid winter wheat varieties in Europe are also genetically structured, and they can be separated into three groups containing accessions from the south, north-west, and north-east of Europe respectively [79]. We could not formally test whether this affected the population structure of the pathogens, as we lacked detailed genetic information about the host on which Bgt samples were collected. However, in Northern Europe there is a clear subdivision between wheat accessions from the west and the east [79], while the pathogen population is uniform across the whole region. Thus, it is unlikely that the population structure of Bgt is strongly affected by that of winter wheat.

Our results on the rate of LD decay (Fig I in S1 Text) and the frequency of mating types (Table A in S1 Text) highlight that sexual reproduction is prevalent in all populations of Bgt in Europe and the Mediterranean. Field observations report frequent chasmothecia formation [80], and during this study we obtained chasmothecia from several countries. This contrasts with previous studies on *Blumeria* that suggested clonal or near-clonal reproduction for both barley and wheat powdery mildew [39,81]. While we did not attempt to estimate the frequency of sexual reproduction, our observations suggest that it is more common than previously proposed.

Additional insights into the population biology of Bgt were obtained by the demographic analysis. We found that the Kingman coalescent is not a suitable model for Bgt populations and that the data is modelled better using the Beta coalescent. The better fit of a multiple merger coalescent model (the Beta coalescent) over the Kingman coalescent indicates that there is a large variability in the reproductive success of different genealogical lineages [53]. In aerially dispersed, obligatory biotrophic plant pathogens, the stochasticity introduced by wind and host availability may cause this increased variance in the offspring distribution. Individuals may produce hundreds of thousands of spores every generation [82] but only a small proportion is successfully dispersed by the wind and lands on a viable host plant to continue reproduction. This is analogous to the sweepstake reproduction of several marine organisms (reviewed in ref. [83]). In this model of "few winners, many losers" [84], a small fraction of individuals can contribute disproportionately more to the next generation, thus increasing variance in reproductive success. Another process that leads to large variation in fitness between different lineages is the recurrence of population bottlenecks [53]. In the case of wheat powdery mildew, these might occur between successive crop growing seasons. In the absence of suitable hosts post-harvest, the pathogen population sizes may fall drastically, with few surviving on green bridges or as sexual spores in chasmothecia. Finally, recurring selective sweeps could also lead to multiple merger genealogies [85], as was observed recently in Atlantic cod populations [86]. In crop pathogens the recurring sweeps may be caused by successive deployment of new resistance genes and fungicides.

Future studies and further method development are needed to test these different hypotheses. Nonetheless, regardless of which of these processes affect Bgt populations, these results have important consequences for how we perform and interpret demographic inference in Bgt, and potentially in other pathogens. Most demographic analyses use the Kingman coalescent without testing its assumption about the variance of reproductive success between lineages. However, ignoring this assumption can lead to the inference of spurious population dynamics such as population growth [54,55]. Beyond Bgt, multiple merger coalescent models have been tested for only one additional agricultural pathogen (*Setosphaeria turcica*, northern corn leaf blight). In that case, the Kingman coalescent resulted to be the best fitting model [87], but this might not be the case for other pathogens with a life cycle similar to Bgt (e.g., mildews and rusts), and researchers working on such pathogens should test whether the Kingman coalescent is an appropriate model prior to demographic analyses.

## Implications for control strategies

Insights into the evolutionary and epidemiological dynamics of crop diseases can be translated into better control methods only if they are on a spatial and temporal scale relevant for agriculture. One of the main results of the population genetic

analysis in this study was the identification and characterization of different Bgt populations. In particular, the identification of one single homogeneous population over Northern Europe has important implications for the deployment of novel resistance genes and suggests that resistance breeding should be coordinated at least among countries in Northern and Eastern Europe. In the future, it might also become possible to leverage epidemiological dynamics for a strategic deployment of major resistance genes. For example, if the hypothesis of asymmetric gene flow in Northern Europe holds true, with higher gene flow from west to east, it would be conceivable to think of a staggered deployment of novel resistance sources starting from the east and moving toward the west; or at least to avoid the opposite pattern, as virulent variants selected on resistant material would move faster in the eastward direction. Whether such strategies are feasible, or cost effective, depends on the magnitude of the asymmetry in gene flow. Such scenarios could be explored with simulation studies, or by using molecular epidemiology to study past cases of resistance breakdown. Importantly, these hypotheses need to be tested in other pathogens as well, as their relevance might extend beyond wheat powdery mildew.

Investigating past examples of resistance genes that have lost effectiveness can provide precious guidance for future strategies. Here we showed that resistance to powdery mildew provided by the wheat receptor Pm17 was lost rapidly because multiple virulent haplotypes were already present in Europe before the introduction of the resistance gene (see also ref. [26] for an analysis based on a global population). Those haplotypes carried variants that were able to partially escape recognition by Pm17 (variants B and C), and in the last two decades they spread throughout the continent. A new virulent variant that is not recognized at all by Pm17 appeared only recently in Northern Europe and Turkey (variant H), and it will be very informative to monitor how it spreads over time and space in the future. Our infection assays on a *Pm3e* (Fig AA in S1 Text) line showed that virulent genotypes are already circulating in natural populations even though *Pm3e* has never been used in breeding. These results are similar to those obtained for oat, as it was reported that British populations of oat powdery mildew contained genotypes that were virulent onto new oat resistant varieties before their commercialization [88]. The existence of virulent genotypes prior to the introduction of new major resistance genes could explain why many of these genes have become ineffective so quickly. Future molecular epidemiology studies of additional *Avr*s in Bgt and other pathogens can reveal if this is indeed a common pattern.

In any case, field trials in one or few countries are likely not sufficient to predict the durability of novel major resistance genes. Complementing this data with infection tests using isolates from broader collections could help identify genes providing durable resistance (for genes that are active at the seedling stage). Such collections should represent the current pathogen population in the region where the resistance gene(s) will be deployed, and their geographic range can be determined by population genetics studies such as ours. For example, for wheat powdery mildew in Northern Europe, new resistant lines should be challenged at least with isolates from the entire range of the Northern European population (N_EUR), and possibly also with isolates from nearby populations in Southern Europe and Turkey.

Finally, as the cost of sampling and sequencing decreases, molecular surveillance of virulence and fungicide resistance at the local scale is becoming feasible. Such data can inform fine-scale decisions about disease control, such as which wheat variety should be sown and what fungicide should be used, based on the local pathogen population.

## Methods

### Sampling

We sampled 276 isolates of *B. g. tritici* across Europe and the Mediterranean in the spring and summer months of 2022 and 2023 (Fig A in S1 Text). A total of 173 isolates were sampled from infected fields of spring, winter, durum and spelt wheat (S1 Data). Of the remaining 103, 96 were collected from susceptible 'trap pots' of young wheat seedlings (8–20 days old) and 7 from infected durum wheat grown in a greenhouse. The samples were collected from over 90 locations spread over 20 countries in Europe and the Mediterranean region, with each sampling site contributing 1–13 samples over two years. Most of the samples (*n* = 254) were propagated from asexual conidiospores while some (*n* = 22) were revived from sexual fruiting bodies (chasmothecia) following the protocol from [46]. The samples were grown in the lab on

Petri dishes containing segments of young wheat leaves placed on an agar-benzimidazole medium. All samples obtained from asexual conidiospores were first propagated on both Kanzler and Inbar which are susceptible cultivars of hexaploid and tetraploid wheat, respectively. This was done to avoid artificially selecting against samples that could infect only wheat of a certain ploidy (e.g., isolates belonging to the *f. sp. dicocci*). However, all samples were found to grow well on the susceptible hexaploid cultivar Kanzler, which was thus chosen as the sole host for all subsequent rounds of propagation. We isolated single colonies to ensure each sample consisted of one genotype, and not a mix of infections. This was achieved by performing a low-density infection of each sample on fresh, 10-day old wheat leaves laid out on Petri dishes filled with the agar-benzimidazole medium. Four days post infection, the Petri dish of each isolate was observed under a binocular microscope and fragments of leaves that housed a single colony were cut, isolated and allowed to grow independently. Once mature, these colonies were propagated further, and the entire process was repeated once again. Each isolate was then propagated until we obtained enough spores (circa 0.05 g) to perform DNA extraction for whole-genome sequencing.

## DNA extraction, sequencing and publicly available data

DNA extraction was performed following a magnetic beads based protocol adapted from [89] for compatibility with the KingFisher Apex 96 System. Following quality checks by gel electrophoresis, Qubit and Nanodrop, whole genomes of all 276 isolates were sequenced to obtain 150 bp paired-end reads with insert sizes approximately 200–350 bp using either the Illumina NovaSeq 6,000 or NovaSeq X Plus instruments, and Illumina Truseq Nano libraries.

We also retrieved publicly available whole genome sequences of 375 *B.g. tritici* isolates collected from around the world between 1980 and 2019, and the sequences of five *B.g. secalis* isolates that were used as outgroups in some downstream analyses (S1 Data) [36,40,45].

The newly generated sequences were combined with the previously available data to be analyzed together. The details of all the isolates used can be found as supporting data (S1 Data).

## Variant calling pipeline

Raw sequence reads were first trimmed based on quality using fastp v0.23.2 [90,91]. Adapters were detected and trimmed based on per-read overlap analysis employing the default settings in fastp. Read quality was assessed for each read in a sliding window manner and subsequent trimming was performed using the operations –cut_front and –cut_right with options cut_front_window_size 1, cut_front_mean_quality 20 and cut_right_window_size 5, cut_right_mean_quality 20, respectively. Overlapping paired-end reads were merged with fastp --merge, with overlap_len_require = 15 and overlap_diff_percent_limit = 10. These merged reads, as well as the unmerged paired-end reads (and unpaired, if any) were mapped separately to the *Blumeria graminis f.sp. tritici* reference genome 96224 [46] using bwa-mem [92]. The reference assembly was updated to include the newly published mitochondrial genome [93] as well as a contig of the mating type absent in the reference genome (Bgt_MAT_1_1_3, described in ref. [39]). The parts of the 'Unknown' chromosome from the older assembly that matched with the mitochondrion were identified with the help of a dotplot [94] and removed. Thus, the final reference assembly (139.3 Mb, available at https://doi.org/10.5281/zenodo.13903934) consisted of 11 chromosomes, the mitochondrial genome, the alternate mating type contig and an unknown chromosome which contained scaffolds that could not be assigned to any chromosome. The alignments produced by bwa-mem were sorted and merged using Samtools v1.17 [95]. Placeholder read-group and library information was added to the alignment files to make them compatible with GATK v.4.4.0 [96], which was used for the subsequent steps of the pipeline. Duplicate reads were marked using GATK MarkDuplicatesSpark. The "coverage" option in Samtools v1.17 was used to calculate the mean coverage for each chromosome from which we derived the genome wide average coverage. Mating types were assigned to all isolates by comparing the coverage over the two alternate mating type genes. Sample-level haplotype calling was performed with GATK HaplotypeCaller with options –ploidy 1 –ERC BP_RESOLUTION to produce a VCF file with calls for each site in the genome. This file was then split by chromosome using bcftools view --regions [95] to facilitate parallelisation in

downstream computation. The steps mentioned up to this point were wrapped in a Python script that took raw fastq files and the reference, along with some quality trimming parameters, as input and generated the per-sample VCF file, as well as summary statistics related to mapping and calling. Details on how the pipeline was called, including the parameter values used, are available at https://github.com/fmenardo/Bgt_popgen_Europe_2024/tree/Bgt_ms.

Samples with average genome-wide coverage less than 15x were excluded from all further analyses ($n$ = 26). The single VCF files of all the remaining samples were merged using GATK CombineGVCFs. The resulting output was then used as input for GATK GenotypeGVCFs which performed joint genotyping on all the samples. The INFO field values were extracted for all variant positions with GATK VariantsToTable and their distribution was visualized in R [97]. This, along with GATK's hard filtering recommendations, was used to inform filtering decisions. Site-level hard filtering was executed using GATK VariantFiltration with filters QD < 10, FS > 55, MQ < 45 and –4 < ReadPosRankSum < 4. Additional sample-level filters were also employed using a custom Python script. For each site in the genome, all sample calls with depth of high quality, informative reads less than 8 (DP < 8) were recoded as missing data. Further, variant calls that were supported by less than 90% of such reads ('heterozygous' calls) were also recoded as missing. The number of occurrences of failure of each of these filters was recorded, and their distributions visualized. Samples which failed the 'heterozygous' filter at more than 200,000 positions (over chromosomes 1–11) and those for which the ratio of the number of variants to the number of heterozygous sites was less than 1 were excluded from further analyses ($n$ = 12).

## Identification of clonal isolates and definition of datasets

We computed a pairwise distance matrix between all individuals based on the number of SNPs between them using the dist.gene function in the R package ape [98] with the options "method = pairwise". The distance between each pair was normalised by the total number of positions compared. We evaluated the distribution of the distances and classified isolates with a genetic distance of less than 9e-05 nucleotide differences per site between them as clones. We identified 35 clonal groups. Thirty-three of these groups contained 2–3 isolates each that were collected from the same pot or field. We retained only one isolate from each such group. The remaining two clonal groups contained isolates that were collected from distant locations but handled together in the laboratory. These groups, containing 9 isolates overall, were excluded completely as they were suspected to be contaminations.

We defined four main datasets using all non-clonal samples of *B.g. tritici* that passed our quality filters, and which we used for subsequent analyses, as follows:

1. *World*: 568 isolates sampled across the world between 1980 and 2023.

2. *Europe+*: 415 isolates sampled across Europe, the Middle East and North Caucasus (between 25°N–60°N and 9°W–60°E).

3. *Europe+_recent*: 368 isolates from *Europe+* that were sampled in or after 2015.

4. *Europe+_2022_2023:* 255 isolates from *Europe+* that were sampled in 2022 and 2023.

## Population structure

**Principal component analysis.** We performed principal component analyses using the glpca method in the R package adegenet [99,100] on two datasets, *World* and *Europe+*. For each dataset, we used all biallelic SNPs from chromosomes 1–11 that were filtered to contain no singletons and less than 10% missing data (1,916,338 and 1,683,143 SNPs, respectively). To check if our results were robust to different filtering criteria, we also performed the PCA on a subset of the *Europe+* SNPs that were subjected to additional, more stringent site-based filters. SNPs with QD < 20 and MQ < 55 were excluded resulting in 1,349,054 biallelic SNPs. The distribution of the proportion of variance explained by

each principal component was visualized and the PC scores of the first three principal components were plotted using ggplot2 [101] in R.

**ADMIXTURE.** We used ADMIXTURE [47] to estimate individual ancestries for each sample in the *World* dataset (568 isolates). Since this model assumes linkage equilibrium among markers, we performed linkage-disequilibrium based pruning of our SNP dataset using PLINK v1.9 [102] with the options –indep-pairwise, window size = 25 kb, step size = 1 SNP and correlation threshold $r^2$ = 0.1. The resulting 156,047 SNPs were used as input for ADMIXTURE and the program was run for 10 replicates for each K value in the range 1–10. The cross-validation errors for each run and value of K were visualized and compared. Ancestry proportions corresponding to the run with the least CV error were visualized for K = 4–9.

**fineSTRUCTURE.** We ran the fineSTRUCTURE analysis on the Europe+ dataset. We selected all biallelic SNPs on the 11 chromosomes for which there was no missing data (1,201,198 SNPs). The local per base recombination rates were obtained from the genetic map produced by Müller and colleagues [46]. Specifically, we calculated the per base recombination rates as the ratio of the genetic distances and the physical distances between markers. We ran fineSTRUCTURE v4.1.0 [48] with default parameters except for the number of iterations in the expectation-maximization algorithm of Chromopainter, which were increased to 50. With these settings the average estimated Ne was 80.9282, and c was estimated to be 0.477647.

The dendrogram inferred by fineSTRUCTURE was used to classify isolates in populations. For different analyses we used different levels of classifications defined from the coarsest, level-1, in which all samples are assigned to two populations, to the finest, level-10, in which fineSTRUCTURE distinguished 45 populations (S1 Data, Fig H in S1 Text).

## Summary statistics

We calculated common population genetics summary statistics for the five populations identified by fineSTRUCTURE level-4 classification (ME, N_EUR, S_EUR1, S_EUR2 and TUR) in the *Europe+_recent* dataset. We note that this dataset has virtually no overlap with the set of isolates used in previous population genomic analyses [40], and the estimates of the summary statistics should therefore not be compared across studies. We computed measures of within-population diversity, namely average per-site nucleotide diversity (pi), Watterson's theta and Tajima's D for each population in windows of 10 kb spanning the whole nuclear genome. To calculate pi, we first generated an all-site VCF for each population, filtered for site-quality and maximum 50% missing data, and then used pixy [103] to obtain pi for all windows. We then filtered the population VCF files to retain only biallelic SNPs with maximum 50% missing genotype calls and used VCFtools SNPdensity [104] to obtain the number of segregating sites (S) in windows of 10 kb across the genome. Watterson's theta was then calculated for each window as $S/a_1$, where $a_1$ is the (n − 1)th harmonic number and $n$ is the sample size. Per-base estimates of Watterson's theta were obtained by dividing $S/a_1$ by the number of valid sites in each window, as reported by pixy. These estimates of pi and Watterson's theta were used to calculate Tajima's D in windows using the formula described in Tajima, 1989 [105]. We also calculated $d_{xy}$ and Weir and Cockerham's $F_{ST}$ in windows of 10 kb using pixy.

Linkage disequilibrium (LD) was calculated using PLINK2 [102,106] for the *Europe+_recent* dataset. All biallelic SNPs were first filtered to retain only SNPs with maximum 10% missing data. Next, separate VCF files were created for the five populations (ME, N_EUR, S_EUR1, S_EUR2 and TUR). We used the options –r2-unphased –ld-window-r2 0 and –ld-window-kb 10 to calculate LD between pairs of SNPs, which was visualized as average $r^2$. Pairs of SNPs that are more than 10 Kb apart were not considered.

## Demographic inference

To avoid the confounding effect of population structure we focused on the finest level of population subdivision (fineSTRUCTURE level-10; Fig H in S1 Text). We selected the four populations with largest sample sizes: N_EUR1, N_EUR2, E_EUR1 and E_EUR2. Furthermore, to avoid potential biases due to clustered and serial sampling we only used isolates sampled from 2022, and we

selected one sample per sampling location. The final dataset was composed of 27, 15, 10 and 17 samples for N_EUR1, N_EUR2, E_EUR1 and E_EUR2, respectively (S1 Data). To identify ancestral and derived alleles we included 5 isolates of the *Blumeria graminis f.s.p secalis* (Bgs), a related form of Bgt which infects rye [45]. We only considered sites that were not polymorphic in Bgs. For polymorphic sites in Bgt, we considered as ancestral the allele carried by Bgs and excluded sites where the allele carried by Bgs was not present in Bgt. Finally, we excluded all genomic sites with missing data, indels, multiallelic SNPs, and multiple site polymorphisms. The remaining sites were used to calculate the genome-wide site frequency spectra and to estimate Tajima's D.

We considered a Beta coalescent with parameter $\alpha$ and exponential growth with rate $g$ (maximizing Eq. (2) in ref. [56], with $e = 0$). This model belongs to a class of models named multiple merger coalescents, which allow for large variation in reproductive success, and consequently more than two lineages can coalesce at a single time point on the coalescent timescale (hence the name multiple mergers). This contrasts with the Kingman coalescent, in which the variance in reproductive success is small enough that at most two lineages can coalesce at a given time point. We used the approximate maximum-likelihood approach from [56]. We maximised the likelihood over a parameter grid with equidistant steps for both parameters. We let $\alpha$ vary from 1 to 2 in steps of 0.01. The size of mergers decreases with increasing $\alpha$, up to no multiple mergers for $\alpha = 2$ (i.e., the Kingman coalescent is a special case of the Beta coalescent corresponding to $\alpha = 2$). We considered exponential growth rates $g$ from 0 (no growth) to 10, in steps of 0.25, where a growth rate of $g$ means that, within one unit of coalescent time, the population grows by $\exp(g)$ (be aware of the difference in timescales between different multiple merger rates, see ref. [56].)

Additionally, we assessed whether adding a chance (parametrized as probability $e$) of having misidentified the derived with the ancestral allele at each SNP affected the results of the inference. For this analysis we also included sites that were monomorphic in Bgs but showed an allele not present in Bgt, as they are used to infer the probability of misidentifying ancestral and derived alleles. Following Freund and colleagues [56], we used a slightly different (composite) approximate likelihood approach (approximate likelihood function given by Eq. (17) in Supplementary Information A.4.1 of [56]). We optimized over a grid of the same $\alpha$ and $g$ values as above, but also probabilities $e$ from 0 to 0.2 in steps of 0.01. The parameter $e$ is estimated both from its effect on the SFS and from a comparison with the outgroup alleles used to call derived alleles in the sample, while considering different mutation rates for transitions and transversion. For this, we additionally optimized a fourth parameter, $\kappa$, that is the ratio between transition and transversion mutation rates (using four values: $\kappa = 1$, the estimate from [56], and 2/3 and 3/2 the values of this estimate). We report the (approximate) likelihood ratio between the best fitting models (with highest likelihoods) with and without a multiple merger component (i.e., $\alpha < 2$ versus $\alpha = 2$). We then graphically assessed the goodness of fit of both the best models with and without multiple merger component by plotting the expected value of the site frequency spectrum divided by the expected number of segregating sites.

### Windscape

We estimated the wind connectivity between different sampling locations using the R package windscape [60]. We used 10 years (2012–2021) of hourly wind data (speed and direction at 10 m above ground) during the main period of dispersal of wheat powdery mildew: February–July [43]. The data was downloaded from the Climate System Forecast Reanalysis [107] dataset ds94.0 (https://doi.org/10.5065/D61C1TXF). Windscape uses the u and v components of wind speed to build a connectivity graph connecting each cell of the grid to its neighbors in proportion to the frequency and speed of the wind over the considered period. Wind distances, defined as the mean estimated time of diffusion between the two locations (in wind hours), were calculated between all pairs of isolates in the *Europe+_recent* dataset.

### Mantel tests

We tested how well the genetic distances between pairs of individuals correlated with the geographic, wind and climatic distances between them using Mantel test, as implemented in the mantel.randtest function of the R package adegenet [99,100].

The genetic distance matrix was the same as the one used to identify clones, as described above. The geographic distance matrix was constructed using the rdist.earth function in the R package fields [108]. The wind distance matrix was obtained from the Windscape analysis. To compute climatic distances between pairs of individuals, we used the climate data CHELSA V2.1 from climatologies 1981–2010 [109,110]. The 19 bioclim variables were complemented with 16 additional BIOCLIM+ variables [111,112] that were chosen based on biological relevance, resulting in a total of 35 climate variables (S2 Data). Information for every sample site was extracted with the coordinates and stacked using the R package raster [113]. To avoid overfitting due to collinearity and multicollinearity, we excluded variables that had absolute pairwise correlation value ≥0.85 with another variable, finally retaining 12 variables (S1 Data). Next, we performed a PCA of these 12 variables using the prcomp function in R and computed the euclidean distance between all pairs of samples based on the first 7 principal components using the 'dist' function in R to obtain a pairwise climatic distance matrix.

We performed the Mantel tests for three datasets: (a) *Europe+_2022_2023*, (b) All samples from 2022−2023 that belonged to the population "N_EUR" (*N_EUR_2022_2023*) and (c) "S_EUR2" (*S_EUR2_2022_2023*), as defined by the level-4 of classification of fineSTRUCTURE. The observed correlation value and *p*-value were reported after 999 permutations of the Mantel test.

We also tested for isolation by geographic distance separately along the east-west and north-south axes for the *N_EUR_2022_2023* and *S_EUR2_2022_2023* datasets. Geographic distance along the north-south axis was computed as $R * d\_lat$, where $R$ = radius of earth (in km) and $d\_lat$ = difference in latitude between pairs of locations (in radians). For distance along the east-west axis, we used $R * d\_long * cosine (mean\_lat)$ with $d\_long$ = difference in longitude between pairs of location (in radians) and $mean\_lat$ = mean latitude of each pair of locations (in radians). The genetic distance matrix and the Mantel test procedure were the same as described above.

## Logistic regression

We tested how well geographic, climatic, and wind distances could explain population structure using logistic regression. Using samples belonging to the *N_EUR_2022_2023* and *S_EUR2_2022_2023* datasets, we modelled which factors could predict whether two individuals belonged to the same or different populations. For each pair of individuals, the response variable (Diff pop) was 0 if they belonged to the same population and 1 if different. The three distance measures were the same as described above. Logistic regression was performed using the lrm function in the R package rms [114] independently for the three variables (**Diff pop ~ geographic distance, Diff pop ~ climatic distance, Diff pop ~ wind distance**) as well as a multiple regression (full model: **Diff pop ~ geographic distance + wind distance + climatic distance**).

## Effective migration surfaces

We estimated effective migration surfaces for the *Europe+_2022_2023* dataset using FEEMS [59]. We filtered out all singletons and missing data and then performed LD-based pruning with PLINK using with the options –indep-pairwise, window size = 25 kb, step size = 1 SNP and correlation threshold $r^2$ = 0.1 The PLINK output files, along with a discrete global grid of triangular cells of a suitable resolution, were used as input for FEEMS. Leave-one-out cross-validation was performed over a range of lambda values (from 1e-6 to 1e2 in steps of 20) and the one with the least cross-validation error was chosen as the most appropriate lambda for the fit (2.06914). The results were plotted using matplotlib [115] in Python.

## Redundancy analysis

We used redundancy analysis to explore what factors shaped patterns of genetic diversity in Bgt in Europe. We tested the effects of local climatic conditions, wind connectivity, geography (sampling location) and country of sampling for the *Europe+_recent* dataset. We used only biallelic SNPs with no missing data, filtered for minor allele frequency 0.05 using

GATK. This data was converted to a binary genotype matrix to be used as response variables for the RDA. We used the 12 selected climatic variables, as described previously (see section "Mantel Tests"). The pairwise wind-distance matrix computed using Windscape (see above) was used to locate each sample in a cartesian space with the R function *cmd-scale*. The first three dimensions (wind coordinates) were used as explanatory variables in the RDA. Forward variable selection for the climatic variables was performed using the 'ordiR2step' function of the R package vegan [116] which selected all 12 of the shortlisted climatic variables. We performed RDA using the function 'rda' from vegan as a full model (**genotypes ~ climate + wind + geography + country**), as well as partial RDAs for each variable, conditioning on all other covariates. The relative contribution of each factor was assessed using ANOVA, as implemented in the 'anova.cca' function of vegan.

We also tested the effect of host ploidy (hexaploid or tetraploid wheat) on genetic variation. For this, we used a subset of the *Europe+_2022_2023* dataset containing isolates that had been sampled from infected fields with known host types (*n* = 131). Genotypes were filtered to include only biallelic SNPs with no missing data and minor allele frequency >0.05. The full RDA model tested was **genotypes ~ climate + wind + geography + country + host**. The covariate data and the following steps of the RDA were the same as described above.

### isoRelate

We used isoRelate [62] to perform genome scans for recent positive selection. We used all the individuals in the dataset *Europe+_recent*, and we performed the analysis separately for the five populations (ME, N_EUR, S_EUR1, S_EUR2 and TUR) corresponding to the fineSTRUCTURE level-4 classification. For each population we selected all SNPs with no missing data and a minor allele frequency greater than 0.05. Additionally, we excluded SNPs that could not be mapped unambiguously on the genetic map. We ran isoRelate to identify "identical-by-descent" (IBDe) segments between pairs of samples and we considered only IBDe segments that were larger than 2 cM, larger than 50 Kb, and with a minimum number of SNPs equal or greater than 50. Two cM correspond to roughly to 25 sexual generations, as the average length in cM of a pair of IBDe segments after *x* generations can be obtained with 100/2*x*. However, we emphasize that the size estimates in cM depend on the accuracy of the genetic map, which was produced by a previous study crossing a wheat powdery mildew isolate with a triticale powdery mildew isolate [46]. In a second step, we used isoRelate to calculate the proportion of IBDe pairs for each SNP and identify SNPs with a significant excess of IBDe pairs. We mapped the location of all known avirulence genes (*AvrPm1.1, AvrPm1.2, AvrPm2, AvrPm3a2/f2, AvrPm3b2/c2, AvrPm3d3, AvrPm8, AvrPm17, AvrPm60*), of the suppressor of virulence (*SvrPm3a1/f1*), and of known fungicide targets (*Btub, cyp51, sdhB, sdhC, sdhD, erg2, erg24*), but for clarity we included in the plots only loci that corresponded to a relatedness peak in at least one population [26,27,30,31,33,34,36,37,117].

### AvrPm17 analysis

**Population genetics and molecular epidemiology of AvrPm17.** We mapped raw reads of samples belonging to the *Europe+* datatset to one single copy of the *AvrPm17* locus (Chr1: 4,365,017−4,365,402 ±2 Kb) and called variants following the previously described variant calling pipeline. Mean read depth over the *AvrPm17* coding sequence was compared to the genome-wide average coverage for each isolate in the *Europe+* dataset to estimate the number of gene copies (coverage ratio between 1.5 and 2.5 means two copies, between 2.5 and 3.5 three copies etc.). The VCF file produced after variant calling and filtering was phased using WhatsHap [118] to differentiate between calls on the two gene copies. We extracted haplotypes for each gene copy from the phased VCF file using bcftools consensus. The resulting haplotypes were translated and classified into protein variants as described in [26]. Protein variants were defined based on the amino acid sequence of the mature protein, i.e., after the removal of the signal peptide (the first 25 amino acids). Previously undescribed variants were named in alphabetic order (variants H to L). We used isoRelate to identify identical-by-descent (IBDe) segments in a region of 3 Mb around the locus of *Avr Pm17* (Chromosome 1 from bp 3,000,to

bp 6,000,000). We analyzed all the isolates together (i.e., without subdividing them in populations), and filtered out sites with missing data or minor allele frequency <0.05. We excluded all IBDe segments shorter than 50 Kb and 2 cM, and with <50 SNPs. We generated clusters by connecting isolates that were IBDe over the region included between the two copies of *AvrPm17* (chromosome 1, from bp 4,365,017 to bp 4,365,017). The relatedness network was visualized with the R package igraph [119,120]. Finally, we inferred the haplotype network of the nucleotide sequence of *Avr Pm17* with the R package pegas, using the parsimony algorithm (haplNet function) [121]. The haplotype count, (i.e., the size of the nodes in the network) corresponds to the number of isolates in which a haplotype was found.

**Plasmid cloning and AvrPm17 – Pm17 co-expression in N. benthamiana.** For *Agrobacterium*-mediated expression in *N. benthamiana, Pm17* and *AvrPm17* variants were cloned into the binary vector pIPKb004 [122]. The expression construct pIPKb004-Pm17-HA has been previously described [26]. The expression constructs pIPKb004-AvrPm17_varA, pIPKb004-AvrPm17_varB, and pIPKb004-AvrPm17_varC, which contain *N. benthamiana* codon-optimized effector sequences lacking the signal peptide, have also been previously described [26]. To generate pIPKb004-AvrPm17_varH, site directed mutagenesis was performed with Phusion HF Polymerase (New England Biolabs) and primer pair LK1123(5′-CCACCGTTCTCAGCC-3′)/LK1124(5′-CACGTATATACCTGCGTCAT-3′), using the gateway compatible entry clone pUC57-AvrPm17_varC [26] as a template resulting in pUC57-AvrPm17_varH. The *AvrPm17_varH* coding sequence was subsequently mobilized into pIPKb004 using LR clonase II (Invitrogen). The pIPKb004-GUS expression construct which served as a negative control has been previously described [123]. All pIPKb004 expression constructs were transformed into *A. tumefaciens* strain GV3101 using freeze-thaw transformation [124].

*Agrobacterium*-mediated co-expression of *AvrPm17* variants and *Pm17* in *N.benthamiana* was achieved using the protocol described in ref. [33]. To do so, *Agrobacteria* were grown overnight in Luria broth (LB) liquid medium supplemented with appropriate antibiotics at 28°C. Prior to infiltration, *Agrobacteria* were harvested by centrifugation (3,300$g$, 7 min) washed once in antibiotic-free LB medium and resuspended in AS medium (10 mM MES-KOH pH 5.6; 10 mM MgCl$_2$; 200 µM Acetosyringone) to a final OD$_{600}$ of 1.2 and subsequentially incubated for 2 h at 28°C for virulence induction. *Agrobacteria* were mixed in a 4:1 (AVR:R) ratio immediately prior to infiltration into *N. benthamiana*. Imaging and quantification of the HR cell-death response was performed 48 h after *Agrobacterium* infiltration with the Fusion FX imaging system (Vilber Lourmat) and the Fiji software as previously described [33].

**Infection tests.**

- **Amigo:** Each isolate was tested on at least five Amigo plants, as well as Nimbus plants as the susceptible control for inoculation effectiveness. Seedlings were inoculated at approximately 10 days old with fully expanded first leaves (DC: 12) [125]. After inoculation, plants were grown at 19 °C/15 °C day/night temperature and a 16hours photoperiod. Host reactions were scored after approximately 8–10 days, once the fungal mycelium was fully developed on the susceptible check. Infection types were indicated according to a 5-level scale [126] where 0, 1, and 2 represented resistant plants (0 means immune, i.e., no visible infection symptoms; 1: hypersensitive reaction with necrotic flecks; 2: small colonies with necrotic flecks, no or scarce sporulation) and 3 and 4 represented susceptible plants (3: moderate mycelial growth and sporulation, small necrotic areas; 4: well-developed mycelium and good sporulation). To prevent powdery mildew contamination, the plants infected by various isolates were grown separately in transparent boxes.

- *Pm17* **transgenic lines:** Virulence on *Pm17* was tested with two previously described transgenic lines (Pm17#34 and Pm17#181), and their sister lines as control [127]. It was shown that these two transgenic lines differ in their Pm17 protein abundance, with Pm#181 being the stronger one. Ten days old leaf fragments were infected and kept in agar plates (0.4% agar and 0.05% benzimidazole). The plates were kept at 20 °C with a 16 h light –8 h dark cycle. Pictures of the infected fragments were taken seven days after infection. Eight isolates collected in 2022 and 2023 were tested in nine replicates. The isolates were selected based on their Avrpm17 protein variants: two isolates each for variants A, B, C and H.

## Infection tests on the Pm3e transgenic line

To test virulence on *Pm3e* we used a transgenic line (line 2) described previously [65]. Its sister line was used as a control. Ten days old leaf fragments were infected and kept in agar plates (0.5% agar and 0.05% benzimidazole). The plates were kept at 20 °C with a 16 h light –8 dark cycle. Pictures of the infected fragments were taken 10 days after infection. We tested 155 isolates collected in 2022 in infection assays performed in three replicates.

## Supporting information

**S1 Text. Tables A–K, Figures A–AA, and Appendices A–C.**
(PDF)

**S1 Data. Sample Metadata.** Metadata associated with all Bgt isolates used in this study.
(XLSX)

**S2 Data. List of BIOCLIM and BIOCLIM+ variables.** Name and description of the bioclimatic variables used in this study.
(XLSX)

## Acknowledgements

We would like to thank Gerhard Herren, Beat Keller, Matthew Kling, Marion Müller, Anne Roulin, Kentaro K. Shimizu, Alexandros G. Sotiropoulos and Helen Zbinden for support and feedback. We would also like to thank all colleagues who contributed to sampling wheat powdery mildew. Part of the genome sequencing was performed at the Functional Genomics Center Zurich (FGCZ) of the University of Zurich and ETH Zurich. All computation work was performed on the Science-Cluster provided by the Science IT team of the University of Zurich. SBo acknowledges support (for sampling) from the Plant Protection Extensions of the Swedish Board of Agriculture (Jordbruksverket).

## Author contributions

**Conceptualization:** Jigisha Jigisha, Fabrizio Menardo.

**Data curation:** Jigisha Jigisha.

**Formal analysis:** Jigisha Jigisha, Jeanine Ly, Nikolaos Minadakis, Fabian Freund, Fabrizio Menardo.

**Funding acquisition:** Fabrizio Menardo.

**Investigation:** Jigisha Jigisha, Jeanine Ly, Lukas Kunz, Urszula Piechota, Fabrizio Menardo.

**Project administration:** Fabrizio Menardo.

**Resources:** Jigisha Jigisha, Urszula Piechota, Beyhan Akin, Virgilio Balmas, Roi Ben-David, Szilvia Bencze, Salim Bourras, Matteo Bozzoli, Otilia Cotuna, Gilles Couleaud, Mónika Cséplő, Paweł Czembor, Francesca Desiderio, Jost Dörnte, Antonín Dreiseitl, Angela Feechan, Agata Gadaleta, Kevin Gauthier, Angelica Giancaspro, Stefania L. Giove, Alain Handley-Cornillet, Amelia Hubbard, George Karaoglanidis, Steven Kildea, Emrah Koc, Žilvinas Liatukas, Marta S. Lopes, Fabio Mascher, Cathal McCabe, Thomas Miedaner, Fernando Martínez-Moreno, Charlotte F. Nellist, Sylwia Okoń, Coraline Praz, Javier Sánchez-Martín, Veronica Sărăţeanu, Philipp Schulz, Nathalie Schwartz, Daniele Seghetta, Ignacio Solís Martel, Agrita Švarta, Stefanos Testempasis, Dolors Villegas, Victoria Widrig, Fabrizio Menardo.

**Software:** Jigisha Jigisha, Jeanine Ly, Nikolaos Minadakis, Fabian Freund, Fabrizio Menardo.

**Visualization:** Jigisha Jigisha, Nikolaos Minadakis, Fabrizio Menardo.

**Writing – original draft:** Jigisha Jigisha, Fabrizio Menardo.

**Writing – review & editing:** Jigisha Jigisha, Jeanine Ly, Nikolaos Minadakis, Fabian Freund, Lukas Kunz, Urszula Piechota, Beyhan Akin, Virgilio Balmas, Roi Ben-David, Szilvia Bencze, Salim Bourras, Matteo Bozzoli, Otilia Cotuna, Gilles Couleaud, Mónika Cséplő, Paweł Czembor, Francesca Desiderio, Jost Dörnte, Antonín Dreiseitl, Angela Feechan, Agata Gadaleta, Kevin Gauthier, Angelica Giancaspro, Stefania Lucia Giove, Alain Handley-Cornillet, Amelia Hubbard, George Karaoglanidis, Steven Kildea, Emrah Koc, Žilvinas Liatukas, Marta S. Lopes, Fabio Mascher, Cathal McCabe, Thomas Miedaner, Fernando Martínez-Moreno, Charlotte F. Nellist, Sylwia Okoń, Coraline Praz, Javier Sánchez-Martín, Veronica Sărăţeanu, Philipp Schulz, Nathalie Schwartz, Daniele Seghetta, Ignacio Solís Martel, Agrita Švarta, Stefanos Testempasis, Dolors Villegas, Victoria Widrig, Fabrizio Menardo.

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
