## [Editor Report · Decision Letter 0]

11 Nov 2024

Dear Dr Menardo, 

Thank you for submitting your manuscript entitled "Population genomics and molecular epidemiology of wheat powdery mildew in Europe" for consideration as a Research Article by PLOS Biology.

Your manuscript has now been evaluated by the PLOS Biology editorial staff, as well as by an academic editor with relevant expertise, and I'm writing to let you know that we would like to send your submission out for external peer review.

Once your full submission is complete, your paper will undergo a series of checks in preparation for peer review. After your manuscript has passed the checks it will be sent out for review. To provide the metadata for your submission, please Login to Editorial Manager (https://www.editorialmanager.com/pbiology) within two working days, i.e. by Nov 13 2024 11:59PM.

Kind regards,

Roli

Roland Roberts, PhD

Senior Editor

PLOS Biology

rroberts@plos.org

---

## [Decision Letter · Decision Letter 1]

2 Jan 2025

Dear Dr Menardo,

Thank you for your patience while your manuscript "Population genomics and molecular epidemiology of wheat powdery mildew in Europe" went through peer-review at PLOS Biology. Your manuscript has now been evaluated by the PLOS Biology editors, an Academic Editor with relevant expertise, and by two independent reviewers. Please accept my apologies for the extra delay incurred over the holiday period.

You'll see that reviewer #1 is broadly positive, but questions some of your assumptions and analytical choices. Reviewer #2 is also positive, but has some textual and presentational suggestions (for example, perhaps the Intro should focus more on questions that are actually answered by the study?), and several of his/her long list of requests may involve additional analyses.

In light of the reviews, which you will find at the end of this email, we are pleased to offer you the opportunity to address the comments from the reviewers in a revision that we anticipate should not take you very long. We will then assess your revised manuscript and your response to the reviewers' comments with our Academic Editor aiming to avoid further rounds of peer-review, although might need to consult with the reviewers, depending on the nature of the revisions.

**IMPORTANT - SUBMITTING YOUR REVISION**

*Resubmission Checklist*

*Published Peer Review*

*PLOS Data Policy*

*Blot and Gel Data Policy*

Sincerely,

Roli Roberts

Roland Roberts, PhD

Senior Editor

PLOS Biology

rroberts@plos.org

REVIEWS:

Reviewer #1: Overall, the manuscript "Population genomics and molecular epidemiology of wheat powdery mildew in Europe" uses genome sequencing data to provide valuable insights into the epidemiological dynamics of Blumeria graminis f.sp tritici in Europe and globally. The study effectively addresses key biological questions and employs a suitable methodology, leading to interesting and credible conclusions.

I am listing below some -rather minor - comments and suggestions aimed at improving the manuscript.

1/ Diversity:

Nucleotide diversity estimates reported in this manuscript are higher than those reported by Storipoulos et al. (ref 37), with a partially similar dataset.

why such differences? the reader must know why

2/ Shared ancestry:

The authors tend to over-interpret shared ancestry as evidence for admixture.

S1 appendix Page 36: shared ancestry is interpreted as admixture. However, shared ancestry is not necessarily caused by admixture. The authors should be more cautious. See figure 2 in Lawson et al. doi: 10.1038/s41467-018-05257-7

L143: Lawson et al. (cited above) show that shared ancestry is not necessarily a hallmark of admixture, especially if the age of samples or the sampling effort are variable: « Because STRUCTURE/ADMIXTURE accounts for the most salient variation, results are greatly affected by sample size in common with other methods. Specifically, groups that contain fewer samples or have undergone little population-specific drift of their own are likely to be fit as mixes of multiple drifted groups, rather than assigned to their own ancestral population. Indeed, if an ancient sample is put into a data set of modern individuals, the ancient sample is typically represented as an admixture of the modern populations, which can happen even if the individual sample is older than the split date of the modern populations and thus cannot be admixed. »

3/ Isolation-by-distance:

It is unclear why the authors didn't use partial Mantel tests, i.e. correlation tests between two matrices, but controlling for the effect of a third distance matrix. 

Fig 2 and corresponding text: the y-axis should be an estimate of identify-by-descent, and it's unclear whether the computed Hamming distance is a good estimate of identify-by-descent. Genetic relatedness (computed with PLINK or Kinship2) is an estimate of identify-by-descent.

See for instance doi: 10.1371/journal.pgen.1006911

L211: please don't spread the (false) idea that small things can travel far, blown by the wind. Many fungal plant pathogens produce very small propagules that do not travel well with wind.

4/ Avirulence gene AvrPm17:

Can the authors comment on the lack of loss-of-function mutations at AvrPm17? This is very surprising. If losing AvrPm17 allows overcoming the resistance conveyed by Pm17, one would expect a higher frequency of loss-of-function mutations.

5/ Tajima's D (L604)

Tajima's D was calculated from Pi, estimated with pixy, and Theta, estimated with VCFtools.

However, VCFtools is notorious for providing wrong estimates of population genetic parameters because it typically assumes that sites with missing data have the same genotype as the reference.

Can the author check whether their estimates of Tajima's D are correct? This could be done, for instance, by computing Tajima's D or Watterson's Theta using the Scitkit-allel package.

6/ Other comments:

L133: the « homogeneity » of the Northern European population is not obvious from S4 and S5 figs. Moreover, the population genetic meaning of « homogeneous » is unclear. Does it mean low diversity? does it mean that most of the variation segregates within populations?

L135: S6 Fig does not show a measure of « relatedness ». In population genetics, relatedness is a measure of the degree of kinship. If the authors want to comment on population size, they should use a proxy for the population size, such as an estimate of the population mutation rate theta.

Reviewer #2:

I have reviewed the manuscript "Population genomics and molecular epidemiology of wheat powdery mildew in Europe" by J. Jigisha and colleagues. Overall, I found this work to be well-structured, addressing clear and biologically important questions with appropriately designed and well-executed analyses. I praise the authors for their careful and comprehensive approach, particularly in documenting the methodology: the analyses and workflows are described in sufficient detail, and all data has been made available. I have some comments and questions that I hope will help improve the quality of this manuscript.

Comments:

- I found that the introduction lays out some very relevant questions such as: "How far can a pathogen disperse in one season?" "Which are the main directions and periods of dispersal?" "How many cycles of sexual and asexual reproduction occur each year?" "Where was the origin of the inoculum initiating a disease outbreak in a field?" "How connected are epidemics in different regions?" (lines 73 - 77). As well as: "1) how large-scale population genomics and molecular epidemiology have become feasible for agricultural pathogens, 2) that they can generate insights about their basic biology, and 3) that they can also provide valuable information for control strategies" (lines 105 - 107). However, due to their complexity and particularity, most of these questions are still unanswered or at most partially answered within the bounds of the wheat powdery mildew in Europe. I propose that the authors revisit the phrasing, so there is a better logic and balance between the main challenges on the field and what they have effectively achieved with their work.

- I found that the introduction effectively highlights several relevant questions, such as: "How far can a pathogen disperse in one season?" "Which are the main directions and periods of dispersal?" "How many cycles of sexual and asexual reproduction occur each year?" "Where was the origin of the inoculum initiating a disease outbreak in a field?" "How connected are epidemics in different regions?" (lines 73-77). They add: "1) how large-scale population genomics and molecular epidemiology have become feasible for agricultural pathogens, 2) that they can generate insights about their basic biology, and 3) that they can also provide valuable information for control strategies" (lines 105-107). However, due to the complexity and specificity of these questions, many remain unanswered or are only partially addressed within the scope of this study on wheat powdery mildew in Europe. I suggest that the authors revisit the phrasing in the introduction to create a clearer balance between the broader challenges in the field and the specific contributions and achievements of their work.

- Lines 118-121: The last sentence of this paragraph already describes the population structure. I suggest starting the next section ("Population Structure in Europe and the Mediterranean") with this description for better flow.

- Figure S2: I did not observe the placement of the Australian samples in the PCA. I assume they are "behind" the USA isolates, as stated in S1 Appendix that they are "indistinguishable from the USA isolates." If this is the case, I suggest plotting some of the Australian points on top to make their placement clear.

- Line 126: While it is evident that geography plays a fundamental role in the population structure, the statement that "all analyses separated the Bgt isolates into five groups" is not entirely accurate. For instance, in the admixture plots, Southern European individuals are differentiated; however, in the PCA analyses, they appear to overlap with other groups.

- Lines 140-141: I believe there is an overinterpretation of the admixture analyses. The authors state: "…While the analysis of population structure could distinguish different populations, we also found evidence for gene flow between them." I disagree. Analyses such as Admixture or fineSTRUCTURE aim to optimize Hardy-Weinberg equilibrium to identify populations in an unsupervised manner. While the coexistence of different ancestry components, or "intermediate genotypes," as described by the authors, may suggest potential gene flow, this requires formal testing. Patterns like these can be explained solely by demography.

- Lines 164-187: I feel there is a lack of a clear hypothesis in the "Demographic inference" section. It should come as no surprise that pathogenic fungi exhibit high variation in reproductive success, making models that account for this, such as the Beta coalescent, more appropriate than the classic Kingman coalescent. While I appreciate the emphasis on the importance of such models, it remains unclear what we have learned from their application to the demographic history of powdery mildew in Europe. Although the analyses indicate "evidence for no or minimal population growth," the observed excess of rare variants across all analyzed populations still requires further interpretation.

- Linked to the previous comment, the authors need to justify the selection of the two populations, N_EUR_2 and E_EUR_2, and show how the SFS (including Tajima's D values) appears for these chosen subpopulations. I wonder whether the results would hold if the sister subpopulation, N_EUR_1, or the entire N_EUR population were chosen instead. Finally, for completeness, the authors may want to consider running the scenario where N_EUR_2 allows for misspecification of the ancestral alleles.

- Lines 246-280: The way this section is written weakens the manuscript. It begins by justifying the sampling strategy of two consecutive years as a means to "investigate the spatial epidemiology." However, after just one paragraph, the authors acknowledge that "with samples from two consecutive years we could not observe a geographic shift in any direction, possibly because of the short time between the two sampling seasons". Given the short duration between sampling events, the lack of a spatio-temporal signal is not unexpected. I suggest that the authors frame the dataset more appropriately, emphasizing that both the older collection and the two-year collection together provide a suitable basis for addressing this epidemiological question.

- Lines 313-314: The presence of gene duplications presents challenges when assessing single haplotypes due to the potential for chimeric combinations. In theory, variant "J" could represent a chimeric mixture of variants "A" and "B." Please ensure that this issue is not affecting your results, particularly for variants with low frequencies.

Discussion: It seems natural to discuss the presence of mountain regions, such as the Alps, as a natural barrier that could restrict gene flow in Southern Europe. Did the authors consider this factor in their analysis?

- Lines 546-551: The authors mention that their SNP filtering strategy was driven by the distribution of their metrics. After inspecting the QD distribution, it is evident that most values range between 20 and 40, yet the filter was set to 10. As a result, this hard filter allows many poor-quality SNPs (with QD values between 10 and 20) into the dataset. A similar issue arises with SNPs having Mapping Quality values < 45. Can the authors provide evidence that the inclusion of these SNPs does not significantly impact the analyses?

Minor comments:

- Lines 61 - 63: Please provide citations.

- SFig20: Consider changing the number of bins in the histogram as it may help to visualise the coverage ratio for those samples where the values are around 2X.

- Line 739: Typo. It should be "two copies" instead of "to copies".

---

## [Editor Report · Decision Letter 2]

21 Feb 2025

Dear Dr Menardo,

Thank you for your patience while we considered your revised manuscript "Population genomics and molecular epidemiology of wheat powdery mildew in Europe" for publication as a Research Article at PLOS Biology. This revised version of your manuscript has been evaluated by the PLOS Biology editors and the Academic Editor.

Based on our Academic Editor's assessment of your revision, we are likely to accept this manuscript for publication, provided you satisfactorily address the following data and other policy-related requests:

a) We wonder if the Title could be more declarative, including an active verb? Maybe change it to something like "Genomic surveillance and molecular epidemiology reveal population dynamics and adaptive evolution of wheat powdery mildew in Europe"

b) The Academic Editor, Sophien Kamoun, thinks that the review process has been particularly informative in this case, and strongly recommends that you opt into our "transparent peer review" option, whereby the decision letter and responses will be published alongside the paper (see the example here: https://journals.plos.org/plosbiology/article/peerReview?id=10.1371/journal.pbio.3002949)

c) Please address my Data Policy requests below; specifically, we need you to supply the numerical values underlying Figs 1BCDEF, 2ABCD, 3ABCD, 4AB, 5ABCD, 6B, S1, S2ABCD, S3, S4AB, S5ABCDEFGH, S6ABC, S7, S9, S10AB, S11AB, S12ABCD, S13, S14ABC, S15, S16ABC, S17AB, S18, S19, S20, S21ABC, S22, S23, S24, either as a supplementary data file or as a permanent DOI’d deposition. Note that if you use Github, because Github depositions can be readily changed or deleted, please make a permanent DOI’d copy (e.g. in Zenodo) and provide this URL (see below).

d) Please cite the location of the data clearly in all relevant main and supplementary Figure legends, e.g. “The data underlying this Figure can be found in S1 Data” or “The data underlying this Figure can be found in https://zenodo.org/records/XXXXXXXX

e) Please make any custom code available, either as a supplementary file or as part of your data deposition. I note that you already provide a Github URL (https://github.com/fmenardo/Bgt_popgen_Europe_2024/tree/Bgt_ms) - as mentioned above, please link to a Zenodo snapshot of this instead.

We expect to receive your revised manuscript within two weeks. 

*Published Peer Review History*

*Press*

Sincerely,

Roli Roberts

Roland Roberts, PhD

Senior Editor

rroberts@plos.org

PLOS Biology

DATA POLICY:

Regardless of the method selected, please ensure that you provide the individual numerical values that underlie the summary data displayed in the following figure panels as they are essential for readers to assess your analysis and to reproduce it: Figs 1BCDEF, 2ABCD, 3ABCD, 4AB, 5ABCD, 6B, S1, S2ABCD, S3, S4AB, S5ABCDEFGH, S6ABC, S7, S9, S10AB, S11AB, S12ABCD, S13, S14ABC, S15, S16ABC, S17AB, S18, S19, S20, S21ABC, S22, S23, S24. NOTE: the numerical data provided should include all replicates AND the way in which the plotted mean and errors were derived (it should not present only the mean/average values).

CODE POLICY

DATA NOT SHOWN?

---

## [Editor Report · Decision Letter 3]

4 Mar 2025

Dear Dr Menardo,

Thank you for the submission of your revised Research Article "Population genomics and molecular epidemiology of wheat powdery mildew in Europe" for publication in PLOS Biology. On behalf of my colleagues and the Academic Editor, Sophien Kamoun, I'm pleased to say that we can in principle accept your manuscript for publication, provided you address any remaining formatting and reporting issues. These will be detailed in an email you should receive within 2-3 business days from our colleagues in the journal operations team; no action is required from you until then. Please note that we will not be able to formally accept your manuscript and schedule it for publication until you have completed any requested changes.

Sincerely, 

Roli Roberts

Senior Editor

PLOS Biology

rroberts@plos.org